# Single-cell-level protein analysis revealing the roles of autoantigen-reactive B lymphocytes in autoimmune disease and the murine model

**Takemichi Fukasawa[1], Ayumi Yoshizaki[1]\*[†], Satoshi Ebata[1], Asako Yoshizaki-Ogawa[1], Yoshihide Asano[1], Atsushi Enomoto[2], Kiyoshi Miyagawa[2], Yutaka Kazoe[3], Kazuma Mawatari[4], Takehiko Kitamori[5], Shinichi Sato[1]\*[†]**

[1]Department of Dermatology, The University of Tokyo Graduate School of Medicine, Tokyo, Japan; [2]Laboratory of Radiology and Biomedical Engineering, The University of Tokyo Graduate School of Medicine, Tokyo, Japan; [3]Department of System Design Engineering, Keio university, Faculty of Science and technology, Tokyo, Japan; [4]Department of Applied Chemistry, The University of Tokyo Graduate School of Engineering, Tokyo, Japan; [5]Department of Mechanical Engineering, The University of Tokyo Graduate School of Engineering, Tokyo, Japan

**\*For correspondence:**
ayuyoshi@me.com (AY);
satos-der@h.u-tokyo.ac.jp (SS)

[†]These authors contributed equally to this work

**Competing interest:** The authors declare that no competing interests exist.

**Abstract** Despite antigen affinity of B cells varying from cell to cell, functional analyses of antigen-reactive B cells on individual B cells are missing due to technical difficulties. Especially in the field of autoimmune diseases, promising pathogenic B cells have not been adequately studied to date because of its rarity. In this study, functions of autoantigen-reactive B cells in autoimmune disease were analyzed at the single-cell level. Since topoisomerase I is a distinct autoantigen, we targeted systemic sclerosis as autoimmune disease. Decreased and increased affinities for topoisomerase I of topoisomerase I-reactive B cells led to anti-inflammatory and pro-inflammatory cytokine production associated with the inhibition and development of fibrosis, which is the major symptom of systemic sclerosis. Furthermore, inhibition of pro-inflammatory cytokine production and increased affinity of topoisomerase I-reactive B cells suppressed fibrosis. These results indicate that autoantigen-reactive B cells contribute to the disease manifestations in autoimmune disease through their antigen affinity.

## Editor's evaluation

By using human samples and the recaptured mouse model, the authors nicely showed the importance of B cell-mediated cytokines in autoimmune diseases.

## Introduction

B lymphocytes have diverse functions in the immune system, such as producing antibodies, presenting antigens, and secreting cytokines (*Yoshizaki et al., 2012*; *Yoshizaki, 2016*). For example, B cells produce interleukin (IL)-6, a pro-inflammatory cytokine, and IL-10, an anti-inflammatory cytokine. B cell populations producing these cytokines are termed as effector B cells and regulatory B cells, respectively, which have the function of inducing, promoting, or inhibiting inflammation, and thereby they are also involved in the differentiation and activation of other immune cells, including helper T

(Th) cells (*Yoshizaki et al., 2012*; *Lund and Randall, 2010*). However, it is still unclear what factors determine the ability of B cells to produce different cytokines.

Such versatile functions of B cells are thought to play a critical role in autoimmune diseases. In fact, rituximab (RTX), an anti-CD20 antibody that eliminates B cells, has shown efficacy in many autoimmune diseases, strongly suggesting that B cells are clinically important in the development of autoimmune diseases (*Randall, 2016*). Systemic sclerosis (SSc), one of the autoimmune diseases, is a poor prognosis disease characterized by three main features: fibrosis in the skin and visceral organs, especially the lung, vasculopathy, and immune abnormalities (*Yoshizaki and Sato, 2015*). Since the pathological mechanisms are not yet understood, a fundamental treatment has not been established. Autoantibodies to nuclear antigens, such as topoisomerase (topo) I and centromere (CENP), are detected in SSc patients even before fibrosis and vasculopathy become apparent (*Burbelo et al., 2019*). Autoantibody specificities are closely associated with SSc subsets: anti-topo I antibody is linked to a severe form of SSc, while anti-CENP Ab is often accompanied by a milder form of SSc (*Kayser and Fritzler, 2015*). Despite the close clinical correlation of autoantibodies in SSc, the pathogenic relationship between autoantibodies and the clinical manifestations of SSc remains unknown because autoantigens are generally intracellular components critical for cell mitosis and there is little evidence to show that autoantibodies are internalized into the cell. The putative autoantigen is inaccessible to these Ab specificities, and therefore, it is unclear how such autoantibodies could contribute to fibrosis. Furthermore, it is completely unknown how the inhibition of cell mitosis-related functions of autoantigens, such as topo I and CENP, by autoantibodies could induce fibrosis. Therefore, autoantibodies are not thought to contribute to tissue damage in SSc (*Sato et al., 2004b*). However, autoreactive B cells themselves are expected to play a role in the SSc pathogenesis, independent of autoantibody production; indeed, improvement of fibrosis following B cell depletion by an anti-CD20 antibody is observed in SSc even before serum autoantibody levels are reduced (*Ebata et al., 2021*). In addition, memory B cells, including autoreactive memory B cells, are abnormally activated in SSc patients (*Tsuchiya et al., 2004*). Studies in mice have also shown that induction of an autoimmune response to topo I causes skin and lung fibrosis characteristic of SSc (*Yoshizaki et al., 2011*; *Maier et al., 2017*). However, in the absence of clear role for autoantibodies in disease progression, we wish to examine whether B cells might play an Ab-independent role.

However, few studies have directly investigated the role of autoantigen-reactive B cells in the pathogenesis of SSc at the protein level. There are three possible reasons for this. First, the number of autoantigen-reactive B cells obtained from patients is low. In general, $10^4$ cells are required to analyze proteins, including cytokines produced by the cells. However, due to the small number of autoantigen-reactive B cells present in vivo, it is difficult to obtain the necessary amount for analysis from blood samples, which are limited in quantity (*Pollmann et al., 2019*). Second, cytokines produced by a single B cell are very small. Although there are few studies examining cytokines produced by each B cell, a large number of studies analyzing $10^4$ B cells indicate that the amount of cytokines produced by a single B cell is assumed to be at the level of fg-ag/ml (*Yoshizaki et al., 2012*; *Pollmann et al., 2019*). Therefore, conventional assays, which have a quantitative limit on pg-fg/ml levels, lack the power to analyze cytokines produced by a single autoantigen-reactive B cell. Moreover, the responsiveness of B cells to antigens is defined by the affinity of the B cell receptor (BCR) to antigens (*Liu et al., 2010*), and autoantigen-reactive B cells may also have the different affinities for an autoantigen via a different degree of somatic hypermutation (*Elliott et al., 2018*). Therefore, it is necessary to analyze autoantigen-reactive B cells at the single-cell level to accurately clarify their functional significance. Third, in order to determine functions of autoantigen-reactive B cells, it is essential to examine cell-to-cell interactions with immune cells, such as T cells. However, the volume of conventional cell culture plates is more than $10^6$ times larger than that of cells, which means that the conventional device size is too large for the reaction field of the small number of cells. Therefore, it is likely that cell-to-cell interactions in vivo have not been adequately simulated ex vivo. For these reasons, the role of autoantigen-reactive B cells in the pathogenesis of autoimmune diseases, including SSc, remains a black box.

In this study, we analyzed the function of autoantigen-reactive B cells from SSc patients and SSc model mice at the single-cell level using the recently established unique single-cell protein analysis device (*Fukasawa et al., 2017*; *Ohashi et al., 2009*) and microspatial cell interaction device (*Nakao et al., 2019a*; *Nakao et al., 2019b*). These novel analyses demonstrate that low and high affinities

for topo I of topo I-reactive B cells were related to anti-inflammatory and pro-inflammatory cytokine production, respectively, which were further associated with the inhibition and development of fibrosis, respectively. Moreover, affinities for topo I also related to helper T (Th) cell differentiation in close contact with T cells. These results indicate that autoantigen-reactive B cells may be involved in the disease manifestations in SSc through their distinct cytokine production and interaction with Th cells.

## Results

### Frequencies of peripheral blood topo I-reactive B cells in anti-topo I antibody-positive SSc patients

In many autoimmune diseases, the CD27$^+$ B cell fraction, memory B cells, is enriched with autoantigen-reactive B cells (*Matsumoto et al., 2014*). CD27$^+$ B cells are also highly activated in SSc (*Sato et al., 2004a*). Therefore, we focused on CD27$^+$ B cells in this study to examine topo I-reactive B cells from human SSc patients. First, to determine the number of topo I-reactive CD27$^+$ B cells in SSc patients, frequencies of topo I-APC$^+$ topo I-PE$^+$ CD27$^+$ CD19$^+$ cells in the peripheral blood were examined with flow cytometric analysis. Their frequencies were significantly higher in anti-topo I antibody-positive SSc patients (n = 111) than in healthy individuals (n = 50) and anti-CENP antibody-positive SSc patients (n = 50; p<0.001; *Figure 1A*). Blocking analysis showed that these signals were specific for topo I (*Figure 1—figure supplement 1*). Moreover, the mean number of topo I-reactive CD27$^+$ B cells in the peripheral blood was 22 ± 21 cells/ml. In the conventional method, $10^4$ B cells are required to determine the concentration of cytokines produced by B cells, which means that 500–50,000 ml of blood is needed to obtain the sufficient number of topo I-reactive CD27$^+$ B cells for the analysis. Therefore, it is difficult to directly examine topo I-reactive CD27$^+$ B cells by conventional methods. In the present study, we used a unique microfluidic-enzyme-linked immunosorbent assay (µELISA) system (*Fukasawa et al., 2017*; *Ohashi et al., 2009*), which enables quantitative analysis of the trace amount of proteins as shown below.

### Topo I-reactive B cells produce a variety of cytokines

To confirm that isolated topo I-APC$^+$ topo I-PE$^+$ CD27$^+$ CD19$^+$ cells produce anti-topo I antibodies, titers of IgG anti-topo I antibodies were examined. Topo I-APC$^+$ topo I-PE$^+$ CD27$^+$ CD19$^+$ cells isolated from the peripheral blood as single cells were cultured in 96-well plates for 48 hr. Then, levels of IgG anti-topo I antibodies and total IgG in the culture medium were examined using the µELISA system (*Figure 1B*). In this study, titers of IgG anti-topo I antibodies were defined as values obtained by dividing OD values of IgG anti-topo I antibodies by total IgG OD values. Titers of IgG anti-topo I antibodies produced by topo I-APC$^-$ topo I-PE$^-$ CD27$^+$ CD19$^+$ cells, which are topo I-non-reactive CD27$^+$ B cells, served as background. We found that only topo I-APC$^+$ topo I-PE$^+$ CD27$^+$ CD19$^+$ cells from anti-topo I antibody-positive SSc patients produced IgG anti-topo I antibodies with an average titer of +6 SD or higher, but these B cells from healthy individuals or anti-CENP antibody-positive SSc patients did not produce IgG anti-topo I antibodies (*Figure 1B*). This result suggests that topo I-APC$^+$ topo I-PE$^+$ CD27$^+$ CD19$^+$ cells from healthy individuals and anti-CENP antibody-positive SSc patients are non-specific topo I-bound B cells, and that topo I-reactive CD27$^+$ B cells are detected only in anti-topo I antibody-positive SSc patients.

To investigate cytokines produced by topo I-reactive B cells, cytokine mRNA expression levels were measured by single-cell PCR. Expression of *IL2*, *IL6*, *IL10*, *IL14*, *IL16*, *IL23*, *IL35,* and *TGFB1* was detected in topo I-reactive CD27$^+$ B cells from anti-topo I antibody-positive SSc patients (*Figure 1C*). Interestingly, most B cells produced mainly one type of cytokines, and only a few B cells produced multiple types of cytokines. In contrast, topo I-non-reactive CD27$^+$ B cells used as a control showed almost no expression of these cytokines. Moreover, both topo I-reactive and topo I-non-reactive B cells stimulated ex vivo produced multiple cytokines, and the difference between them found under unstimulated conditions was lost (*Figure 1D*).

These results suggest that topo I-reactive CD27$^+$ B cells are already stimulated by self-antigens in vivo, and that this affects the difference in cytokine profiles between topo I-reactive B cells and non-reactive B cells under unstimulated conditions.

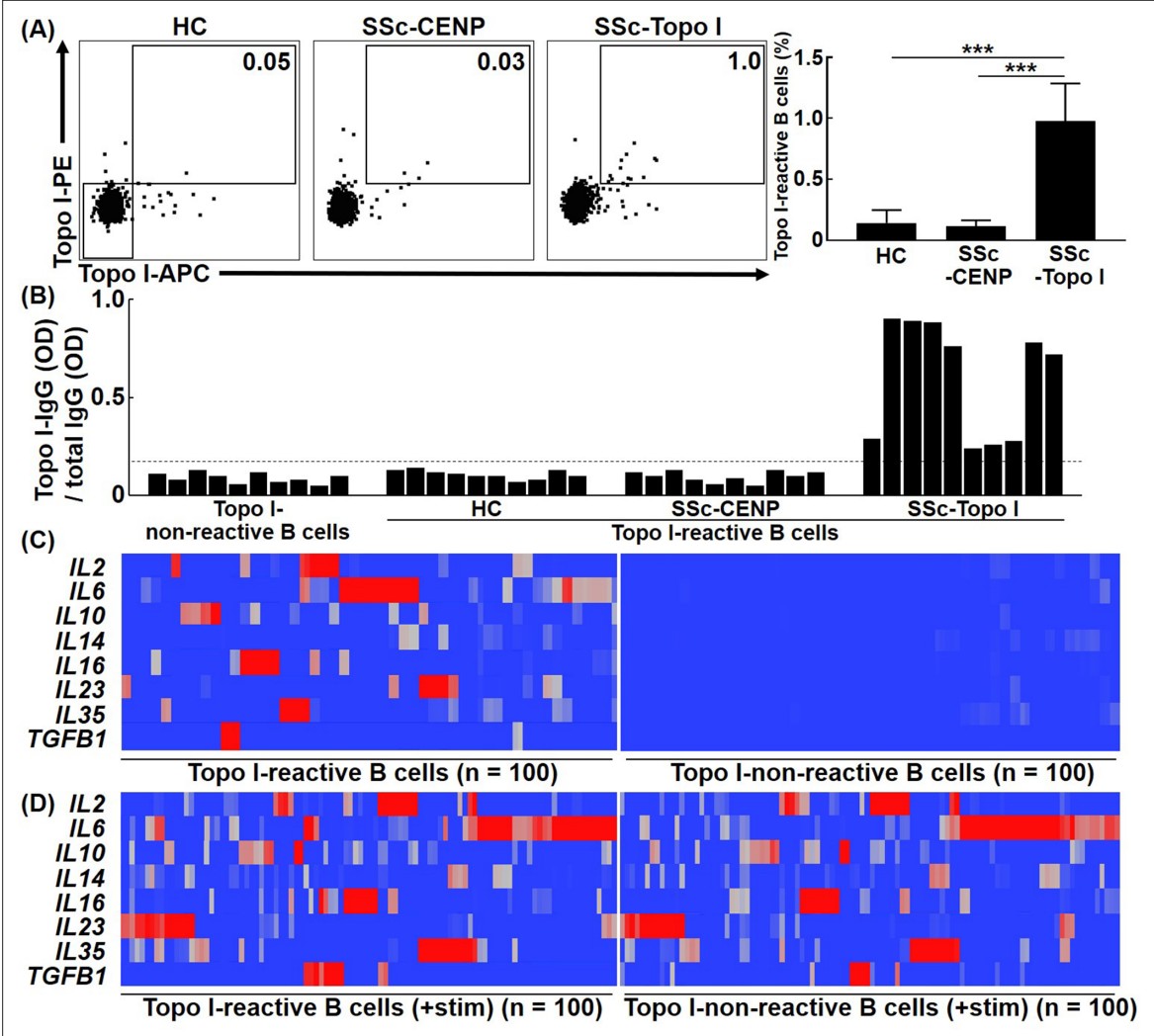

**Figure 1.** Frequencies and cytokine production ability of topo I-reactive B cells in systemic sclerosis (SSc) patients. (**A**) Frequencies of topo I-reactive cells in CD27[+] CD19[+] cells obtained from the peripheral blood of 50 healthy controls (HC), 50 anti-CENP antibody-positive SSc patients, and 111 anti-topo I antibody-positive SSc patients were examined with flow cytometric analysis. Topo I-PE[+] topo I-APC[+] cells were identified as topo I-reactive B cells. Topo I-PE[-] topo I-APC[-] cells were used as topo I-non-reactive B cells in the following experiments. The bar graphs show the mean + SD. ***p<0.005. (**B**) Topo I-reactive B cells were isolated using a cell sorter and subjected to single-cell culture for 48 hr. Subsequently, levels of IgG anti-topo I antibody and total IgG in the supernatant were measured using the μELISA system. The IgG anti-topo I antibody titer produced by each B cell was determined by dividing the IgG anti-topo I antibody OD value by the total IgG OD value. Data for 10 cells in each group are represented. The dotted line represents IgG anti-topo I antibodies with an average titer of +6 SD. (**C**) Real-time-RT-PCR for cytokines was performed at the single-cell level using topo I-reactive B cells and topo I-non-reactive B cells, 100 each, obtained from anti-topo I antibody-positive SSc patients, and is shown in the heatmap. The color of the hea map indicates the degree of mRNA expressions, which are higher as the color of the heatmap changes from blue to red. (**D**) Upon stimulation with PMA and ionomycin, both topoI-reactive CD27[+] B cells and topo I-non-reactive CD27[+] B cells produced a variety of cytokines and showed similar cytokine profiles.

The online version of this article includes the following figure supplement(s) for figure 1:

**Source data 1.** Source file for frequencies and cytokine production ability of topo I-reactive B cells in systemic sclerosis (SSc) patients.

**Figure supplement 1.** Blocking effect of topo I to frequencies of topo I-reactive B cells in anti-topo I antibody-positive systemic sclerosis (SSc) patients.

## The affinity of B cells for topo I regulates their cytokine production capacity

Typically, B cells exposed to an antigen in vivo will increase their affinity for the antigen by somatic hypermutation until the antigen is eliminated (*Meffre et al., 2001*). Ongoing somatic hypermutation is observed for autoantibodies in human autoimmune diseases (*Elliott et al., 2018*). For topo I-reactive

B cells, the affinity for topo I is also expected to increase over the disease course. Therefore, we hypothesized that different affinities for topo I affect the cytokine production ability of topo I-reactive B cells. To this end, topo I-reactive CD27$^+$ B cells (200 cells from 58 anti-topo I antibody-positive SSc patients) were divided into the low-affinity group (80 cells) and high-affinity group (120 cells), according to IgG anti-topo I antibody titers (*Figure 2A*): titers of IgG anti-topo I antibodies in the low-affinity group were 2–4 SD above the background, whereas titers of the high-affinity group were 6–8 SD above it. The high-affinity group has actually low Kd value (high affinity) and the low-affinity group has high Kd value (*Figure 2—figure supplement 1*). Almost all of these cells were CD38$^-$CD95$^+$ cells (*Figure 2—figure supplement 2*), and expression of co-stimulatory molecules (CD80/86, ICOSL, and CD40) in topo I-non-reactive B cells and topo I-reactive B cells (topo I-PE$^+$Topo I-APC$^+$CD19$^+$CD27$^+$) with low or high affinity for topo I in anti-topo I antibody-positive SSc patients was elevated (*Figure 2—figure supplement 3*). The production of IL-2, IL-6, IL-23, and TGF-β1 as well as anti-inflammatory cytokines, IL-10 and IL-35, was examined at the single-cell level in these two groups (*Figure 2B*). The low-affinity group showed higher frequencies of B cells producing anti-inflammatory cytokines, IL-10 and IL-35, while the high-affinity group exhibited higher frequencies of B cells producing pro-inflammatory cytokines, IL-6 and IL-23. On the other hand, frequencies of B cells producing IL-2 and TGF-β1 did not significantly differ between the two groups. Furthermore, the protein expression of cytokines by IL-2, IL-6, IL-10, IL-23, IL-35, and TGF-β1-producing B cells was measured at the single-cell level (*Figure 2C*). The production of IL-6 and IL-23 was significantly higher in the high-affinity group than in the low-affinity group (p<0.005, respectively); conversely, the low-affinity group exhibited significantly higher production of IL-10 and IL-35 relative to the high-affinity group (p<0.005, respectively). There was no significant difference in the amount of IL-2 and TGF-β1 production between the two groups. Thus, topo I-reactive B cells with high affinity for topo I predominantly produced pro-inflammatory cytokines, IL-6 and IL-23, whereas those with low affinity for topo I mainly produced anti-inflammatory cytokines, IL-10 and IL-35. μELISA system revealed relationships between cytokine profiles, autoantigen affinity, and intracellular kinase activity. In IL-10-producing B cells, intracellular signaling and NF-kB signaling were reduced, and were mediated by p38 kinase, Jak1, Stat3, and ATF1 transcription factor. Similarly, IL-35-producing regulatory B cells were related to EOLA1 activation, which inhibits IL-6 signaling, HPK1, which terminate signaling cascade, and Fos transcription factor. IL-6-producing effector B cells were associated with themis2 and HCST, which strengthen signaling pathway, BRAF, KICSTOR complex, SYK, FYN, STAT1, and NFkB2 transcription factor. IL-23-producing B cells were associated with RAF1, JKAMP, and CRTC3 transcription factors (*Figure 2D*).

## Topo I-reactive B cells regulate T cell differentiation via cytokine production

The finding that topo I-reactive CD27$^+$ B cells produced a variety of cytokines suggests that they may control T cell functions via cytokine production. Therefore, we examined the effect of topo I-reactive CD27$^+$ B cells on the differentiation of Th cells. Topo I-reactive CD27$^+$ B cells with low affinity and high affinity for topo I (100 cells each) and control topo I-non-reactive CD27$^+$ B cells (100 cells) were determined from the peripheral blood of anti-topo I antibody-positive SSc patients and co-cultured with 10$^4$ CD4$^+$ T cells on a 96-well plate for 48 hr. Subsequently, *FOXP3* and *RORC* mRNA expression in CD4$^+$ T cells was examined by real-time RT-PCR (*Figure 3A*); however, none of the CD27$^+$ B cells affected *FOXP3* and *RORC* expression using the 96-well plate. We hypothesized that the reason why topo I-reactive CD27$^+$ B cells did not affect T cells was that the 96-well plate could not mimic the size of the reaction field in vivo, that is, cytokines produced by B cells could diffuse before fully acting on T cells. Therefore, we used the unique microculture plate (*Figure 3B*), which enables cell culture in a microspace, and cultured the cells in the same way as the 96-well plate. The volume of culture medium is 200 μl for the 96-well plate, whereas 1 μl is sufficient for the microculture plate. In co-culture with the microculture plates, CD27$^+$ B cells with low affinity for topo I upregulated *FOXP3* expression in CD4$^+$ T cells, while CD27$^+$ B cells with high affinity for topo I upregulated *RORC* expression (*Figure 3C*). Upregulated Foxp3 and RORγt protein expression in these CD4$^+$ T cells was also confirmed by fluorescent cell staining (*Figure 3D*). Furthermore, neutralizing antibodies against IL-10 or IL-35 inhibited Foxp3 expression in CD4$^+$ T cells, while anti-IL-6 or IL-23 antibodies suppressed RORγt expression (*Figure 3E*). Thus, topo I-reactive B cells induced Th cell differentiation in close contact with T cells,

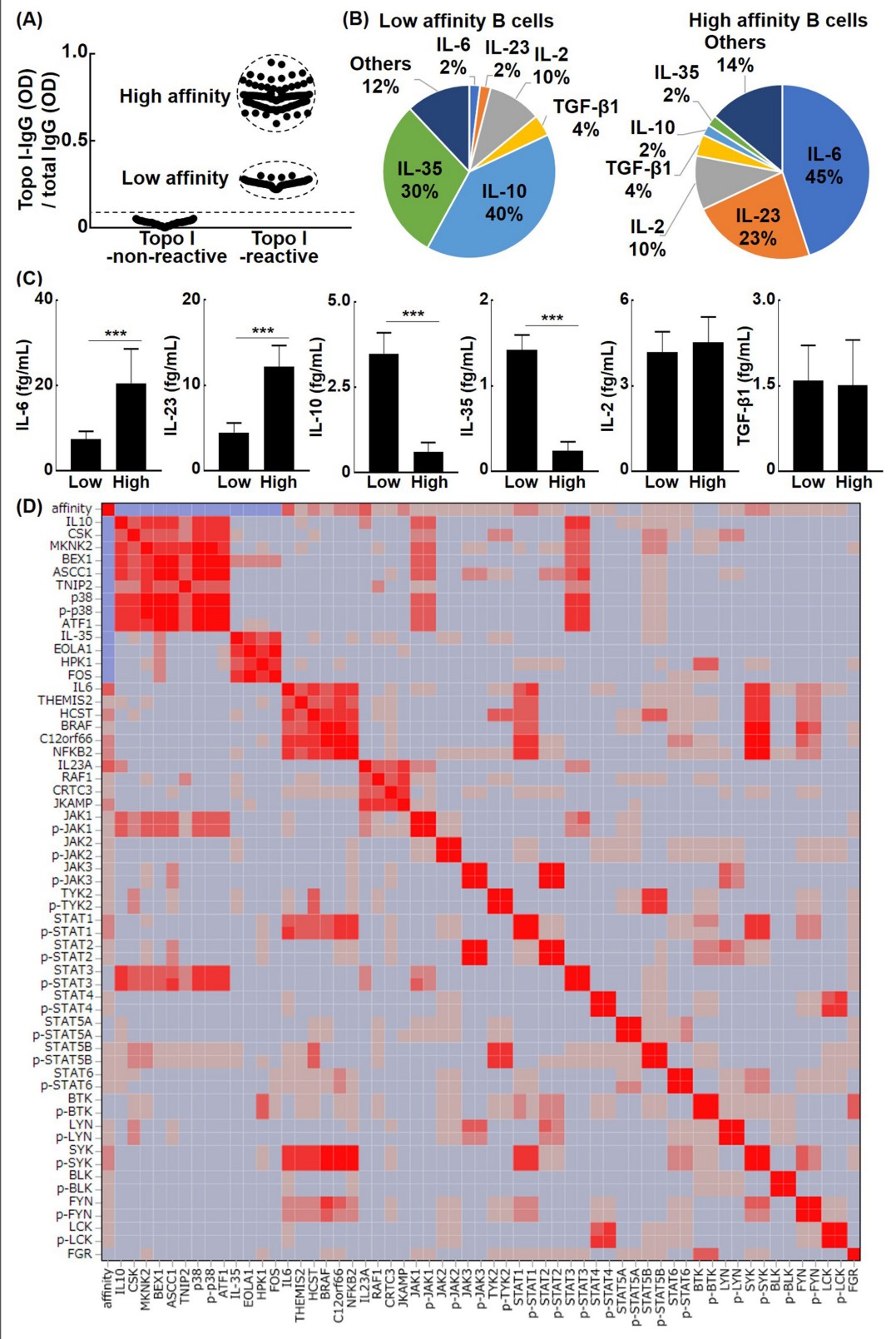

**Figure 2.** The relationship between the affinity for topo I in topo I-reactive B cells and their ability to produce cytokines in systemic sclerosis (SSc) patients. (**A**) Topo I-non-reactive B cells and topo I-reactive B cells were isolated from the peripheral blood of anti-topo I antibody-positive SSc patients, and topo I titers were measured at the single B cell level. Topo I-reactive B cells were divided into two groups: B cells with low or high affinity for

*Figure 2 continued on next page*

*Figure 2 continued*

topo I. (**B**) The frequency of B cells producing each cytokine is represented in the pie charts. (**C**) The amount of each cytokine produced by these cytokine-producing B cells was measured. For these experiments, each of the 300 topo I-reactive B cells and topo I-non-reactive B cells from 111 anti-topo I antibody-positive SSc patients were used. (**D**) Affinity for topo I, cytokines, and intracellular proteins was analyzed at the single-cell level using topo I-reactive B cells, 100 each, obtained from anti-topo I antibody-positive SSc patients, and correlation heatmap is shown. μELISA was used to measure the cytokines, affinity, intracellular kinase, and phosphorylated kinase in each single cell. The color of the heatmap indicates the degree of correlations, which are higher as the color of the heatmap changes from blue to red. The bar graphs show the mean + SD. ***p<0.005.

The online version of this article includes the following figure supplement(s) for figure 2:

**Source data 1.** Source file for the relationship between the affinity for topo I in topo I-reactive B cells and their ability to produce cytokines in systemic sclerosis (SSc) patients.

**Figure supplement 1.** Affinity analysis and correlation of affinities for topo I in topo I-reactive B cells with topo I-IgG(OD)/total IgG(OD).

**Figure supplement 1—source data 1.** Source file for affinity analysis and correlation of affinities for topo I in topo I-reactive B cells with topo I-IgG(OD)/total IgG(OD).

**Figure supplement 2.** Frequencies of CD38-CD95+ cells in topo I-PE+Topo I-APC+CD19+CD27+ B cells in anti-topo I antibody-positive systemic sclerosis (SSc) patients.

**Figure supplement 2—source data 1.** Source file for frequencies of CD38-CD95+ cells in topo I-APC+CD19+CD27+ B cells in anti-topo I antibody-positive systemic sclerosis (SSc) patients.

**Figure supplement 3.** Expression of co-stimulatory molecules in topo I-PE+Topo I-APC+CD19+CD27+ B cells in anti-topo I antibody-positive systemic sclerosis (SSc) patients.

**Figure supplement 3—source data 1.** Source file for expression of co-stimulatory molecules in topo I-PE+Topo I-APC+CD19+CD27+ B cells in anti-topo I antibody-positive systemic sclerosis (SSc) patients.

and the ability to induce Th cell differentiation depended on the cytokine production profile determined by differences in the affinity for topo I.

## Increased affinity to topo I of topo I-reactive B cells is associated with the development of fibrosis and distinct cytokine production in the topo I-induced SSc model mice

To investigate the impact of the increased affinity for topo I of topo I-reactive B cells on the development of fibrosis, we used our recently established topo I-induced SSc model mice (*Yoshizaki et al., 2011*; *Figure 4A*). Mice immunized four times with topo I induced skin and lung fibrosis, whereas those immunized only once did not show skin and lung fibrosis (*Figure 4B*). The frequency of topo I-APC+ topo I-PE+ CD19+ cells in splenic B cells was significantly increased in mice immunized both once and four times with topo I compared to non-immunized mice (p<0.01; *Figure 4C and D*), and the increase was more pronounced in mice immunized four times than in those immunized once (p<0.05). Further analysis at the single-cell level showed that titers of IgG anti-topo I antibodies produced by topo I-APC+ topo I-PE+ CD19+ cells (100 cells) from mice immunized four times with topo I were significantly increased compared to those immunized once (100 cells; p<0.01; *Figure 4E*). Cytokines produced by each of the topo I-APC+ topo I-PE+ CD19+ cells were examined at the single-cell level. The topo I-APC+ topo I-PE+ CD19+ cell population from mice immunized once with topo I predominantly had cells producing inhibitory cytokines, IL-10 or IL-35 (71%), whereas that from mice immunized four times mainly had cells producing IL-6 or IL-23 (68%; *Figure 4F*). Levels of cytokine proteins by these topo I-APC+ topo I-PE+ CD19+ cells were measured at the single-cell level, showing that IL-10 and IL-35 levels were significantly higher in mice immunized once with topo I (p<0.005), while IL-6 and IL-23 levels significantly increased in mice immunized four times (p<0.005; *Figure 4G*). There was no difference in frequencies of IL-2 and TGF-β1-producing B cells and the amount of IL-2 and TGF-β1 between mice immunized once and those immunized four times (*Figure 4F and G*). Thus, as immunization with topo I was repeated, the topo I affinity of topo I-reactive B cells increased, leading to the development of fibrosis with the cytokines produced by topo I-reactive B cells shifted from inhibitory cytokines, IL-10 and IL-35, to pro-inflammatory cytokines, IL-6 and IL-23. We also examined the relationship between the affinity for topo I in topo I-reactive B cells and their ability to produce cytokines

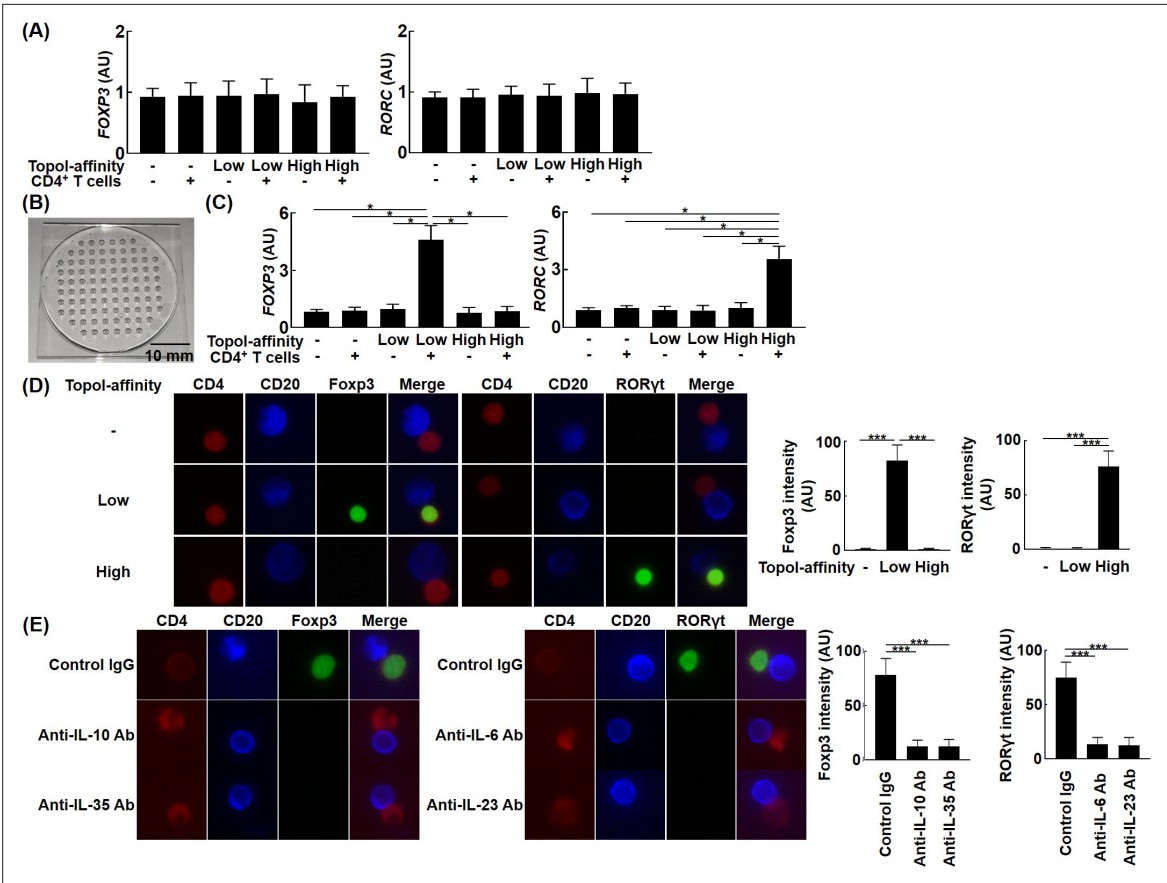

**Figure 3.** Effects of topo I-reactive B cells on the differentiation of CD4+ T cells in systemic sclerosis (SSc) patients. B cells with low affinity for topo I and those with high affinity for topo I as well as topo I-non-reactive B cells were obtained from anti-topo I antibody-positive SSc patients (n = 111). These B cells were co-cultured with CD4+ T cells. After 48 hr of co-culture, mRNA was extracted from these cells and FoxP3 and ROR γ t expression levels were examined by real-time RT-PCR. The results were presented when 96-well plates were used as a co-culture site (**A**) and when microculture plates (**B**) were used (**C**). These cells were further co-cultured on microculture plates, and the protein expression of CD4, CD20, FoxP3, and ROR γ t was confirmed by fluorescent cell staining and signal intensity was determined by ImageJ (**D**). Similarly, co-culture in microculture plates in the presence of anti-IL-10 (10 μg/ml), anti-IL-35 (5 μg/ml), anti-IL-6 (1 μg/ml), or anti-IL-23 (5 μg/ml) antibodies (Abs) was conducted, followed by fluorescent cell staining (**E**). These results represented seven experiments. The bar graphs show the mean + SD. Original magnification, ×1000. *p<0.05.

The online version of this article includes the following figure supplement(s) for figure 3:

**Source data 1.** Source file for the effects of topo I-reactive B cells on the differentiation of CD4+ T cells in systemic sclerosis (SSc) patients.

in complete SSc model mice (*Figure 4—figure supplement 1*). The same results were obtained and topo I-reactive CD19+ B cells were CD38-CD95+, which were activated B cells, or germinal center B cells.

## Topo I-reactive B cells with high affinity for topo I are associated with the disease severity of human SSc

Since the results so far suggest that the cytokine profiles produced by topo I-reactive CD27+ B cells differ according to their affinity for topo I, we examined the relationship of their affinity for topo I with the disease severity in anti-topo I antibody-positive SSc patients (n = 30; *Figure 5*). The ratio of high-affinity CD27+ B cells to low-affinity CD27+ B cells correlated positively with the modified Rodnan total skin thickness score (mRTSS), a semiquantitative measure of skin sclerosis (p<0.0001), and inversely with the percent predicted values of forced vital capacity (%FVC; p<0.0001) and diffusion capacity of the lung for carbon monoxide (%DLco; p<0.0001). In addition, we examined clinical correlation of the affinity for topo I by dividing SSc patients into two groups, the high-affinity B cell-dominant group (n = 58) and the low-affinity B cell-dominant group (n = 53): the former had a higher frequency (>50%) of B cells with high affinity for topo I among CD27+ B cells, while the latter had a higher frequency (>50%)

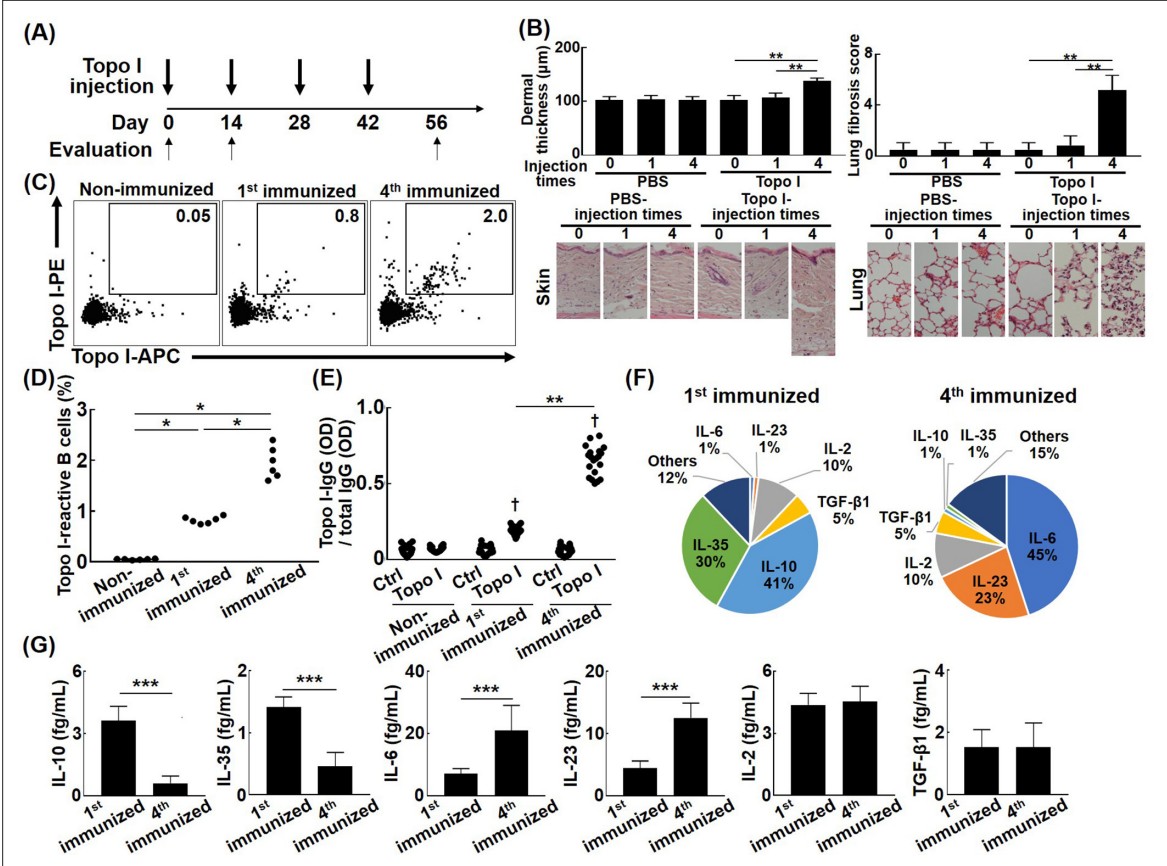

**Figure 4.** The development of fibrosis and cytokine production by topo I-reactive B cells in the topo I-induced systemic sclerosis (SSc) model mice. (**A**) Topo I-induced SSc model mice were generated by immunizing topo I up to four times every 2 weeks. At days 0, 14, and 56, mice were sacrificed and used for study as non-immunized, first immunized, and fourth immunized mice, respectively. Mice treated with phosphate-buffered saline (PBS) instead of topo I were used as controls. Six mice in each group were used. (**B**) Dermal thickness and lung fibrosis score, which reflect the extent of skin and lung fibrosis, respectively, were examined histologically using skin (original magnification, ×40) and lung tissues (×100). (**C**) Topo I-reactive B cells in splenic B cells from these mice were identified with flow cytometric analysis. Frequencies of topo I-reactive B cells in total B cells (**D**) and the IgG anti-topo I antibody titer in each of the topo I-reactive B cells (topo I) and topo I-non-reactive B cells (Ctrl) are shown (**E**). Each of the 100 topo I-reactive B cells isolated from mice immunized once and four times with topo I was analyzed. Frequencies of B cells producing cytokines (**F**) and the amount of produced cytokines are shown in the pie charts (**G**). The bar graphs show the mean + SD. *p<0.05, **p<0.01, ***p<0.005. †p<0.001 vs. each Ctrl.

The online version of this article includes the following figure supplement(s) for figure 4:

**Source data 1.** Source file for the development of fibrosis and cytokine production by topo I-reactive B cells in the topo I-induced systemic sclerosis (SSc) model mice.

**Figure supplement 1.** The relationship between the affinity for topo I in topo I-reactive B cells and their ability to produce cytokines in complete systemic sclerosis (SSc) model mice.

**Figure supplement 1—source data 1.** Source file for the relationship between the affinity for topo I in topo I-reactive B cells and their ability to produce cytokines in complete systemic sclerosis (SSc) model mice.

of B cells with low affinity for topo I (**Table 1**). Patients in the high-affinity-dominant group exhibited longer disease duration (p<0.001), higher mRTSS (p<0.001), more frequent digital scars or ulcers (p<0.01), and lower values of %FVC (p<0.001) and %DLco (p<0.001) compared to the low-affinity B cell-dominant group. Thus, affinities for topo I of topo I-reactive B cells were significantly correlated with the severity of skin and lung fibrosis in SSc patients.

## Topo I-reactive B cells inhibit or induce fibrosis according to differences in their affinity to topo I

To further investigate the pathogenetic relevance of different affinities of topo I-reactive B cells, adoptive transfer experiments were performed using topo I-APC⁺ topo I-PE⁺ CD19⁺ cells isolated from mice

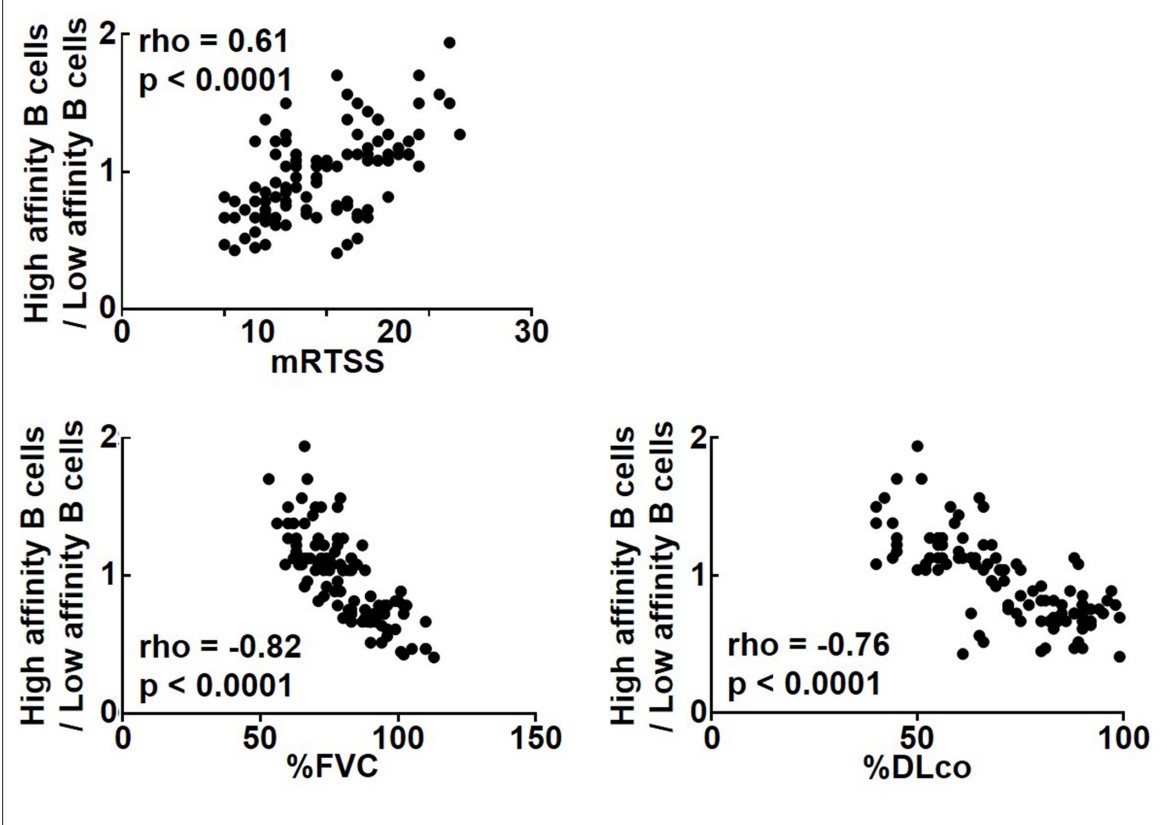

**Figure 5.** Correlation of affinities for topo I in topo I-reactive B cell with clinical parameters for skin and lung fibrosis in systemic sclerosis (SSc) patients. In anti-topo I antibody-positive SSc patients (n = 111), the ratio of B cells with high affinity for topo I (high-affinity B cells) to B cells with low affinity for topo I (low-affinity B cells) was correlated with modified Rodnan total skin thickness score (mRTSS), percent predicted values of forced vital capacity (%FVC), and percent predicted values of diffusion capacity of the lung for carbon monoxide (%DLco).

The online version of this article includes the following figure supplement(s) for figure 5:

**Source data 1.** Source file for correlation of affinities for topo I in topo I-reactive B cell with clinical parameters for skin and lung fibrosis in systemic sclerosis (SSc) patients.

after one or four immunizations with topo I. Wild-type mice adopted with these B cells were immunized three times with topo I (**Figure 6A**). Adoptive transfer of topo I-APC+ topo I-PE+ CD19+ cells from mice immunized once significantly inhibited skin and lung fibrosis compared to non-adoptively transferred mice (**Figure 6B**; p<0.05). Same results were obtained in incomplete topo I model, generated by immunizing topo I for one time, followed by adjuvant immunization up to three times every 2 weeks (**Figure 6—figure supplement 1**). By contrast, topo I-APC+ topo I-PE+ CD19+ cells obtained after four immunizations significantly enhanced skin and lung fibrosis (p<0.05). Remarkably, adoptive transfer of topo I-APC+ topo I-PE+ CD19+ cells obtained after four immunizations induced skin and lung fibrosis even in wild-type mice not immunized with topo I (**Figure 6C and D**). Skin and lung fibrosis was abrogated by administration of neutralizing antibodies against IL-6 or IL-23 (**Figure 6E**). Similarly, adoptive transfer of topo I-APC+ topo I-PE+ CD19+ cells obtained from IL-6 or IL-23-deficient mice immunized with topo I four times into non-immunized wild-type mice did not induce skin and lung fibrosis (**Figure 6F**). Adoptive transfer experiments of B cell-specific Blimp1 CKO mice (PRDM1 CKO) induced skin sclerosis and lung fibrosis. On the other hand, adoptive transfer experiments of B cell-specific class II CKO mice (class II CKO) did not induce skin thickening or lung fibrosis. These results suggest that the B-T interaction is important in this model (**Figure 6—figure supplement 2**). In this model, it was thought that B cells activated T cells, causing skin fibrosis and lung fibrosis. Collectively, topo I-reactive B cells suppressed fibrosis when their affinity for topo I was low, while they induced or exacerbated fibrosis if their affinity was high, and that these functions were exerted through distinct cytokine production by topo I-reactive B cells.

**Table 1.** Clinical correlation of the affinity for topo I of topo I-reactive B cells in anti-topo I antibody-positive SSc patients.

| Characteristics | High-affinity B cell-dominant group* | Low-affinity B cell-dominant group* | p-Value |
|---|---|---|---|
| Sex, number of males/females | 4/54 | 5/48 | N.S. |
| Age at onset, mean ± SD (age) | 52 ± 9 | 49 ± 7 | N.S. |
| Disease duration, mean ± SD (years) | 5 ± 2 | 2 ± 1 | <0.001 |
| Number with dcSSc/lcSSc | 50/8 | 47/6 | N.S. |
| *Clinical features* | | | |
| mRTSS, mean ± SD | 23 ± 5 | 16 ± 4 | <0.001 |
| Pitting scars or ulcers (%) | 62 | 13 | <0.01 |
| *Lung involvement* | | | |
| Interstitial lung disease (%) | 78 | 68 | N.S. |
| %FVC, mean ± SD (%) | 70 ± 9 | 91 ± 11 | <0.001 |
| %DLco, mean ± SD (%) | 57 ± 11 | 83 ± 10 | <0.001 |
| *Laboratory findings* | | | |
| Serum IgG levels, mean ± SD (mg/dl) | 1512 ± 371 | 1398 ± 353 | N.S. |
| Serum IgM levels, mean ± SD (mg/dl) | 159 ± 16 | 179 ± 22 | N.S. |
| CRP levels, mean ± SD (mg/dl) | 0.28 ± 0.19 | 0.21 ± 0.14 | N.S. |
| Anti-topo I antibody levels, mean ± SD (U/ml) | 144 ± 32 | 147 ± 43 | N.S. |

Unless noted otherwise, values are the number of observations.

*The high-affinity B cell-dominant group had a higher frequency (>50%) of B cells with high affinity for topo I among CD27+ B cells, while the low-affinity B cell-dominant group had a higher frequency (>50%) of B cells with low affinity for topo I.

dcSSc, diffuse cutaneous systemic sclerosis; lcSSc, limited cutaneous systemic sclerosis; CRP, C-reactive protein; N.S., not significant; mRTSS, modified Rodnan total skin thickness score; %FVC, percent predicted values of forced vital capacity; %DLco, percent predicted values of diffusion capacity of the lung for carbon monoxide; SSC, systemic sclerosis.

## Bruton's tyrosine kinase (BTK) induces the development and affinity maturation of topo I-reactive B cells

Affinity maturation of BCR by somatic hypermutation results from activation of activation-induced cytidine deaminase (AID) by antigenic stimulation and co-stimulation to BCR (*Pone et al., 2012*). BTK plays an important role in intracellular signal transduction pathways from BCR stimulation to AID activation, and BTK inhibitors suppress somatic hypermutation of BCR by indirectly inhibiting AID activation (*Halcomb et al., 2008*; *Morande et al., 2019*). When ibrutinib, a BTK inhibitor, was administrated in the topo I-induced SSc model mice generated by four times immunization with topo I, frequencies of topo I-APC+ topo I-PE+ CD19+ cells were significantly reduced, and skin and lung fibrosis was also attenuated compared to the model mice without the BTK inhibitor (p<0.05, respectively; *Figure 7A and B*). There was no difference in serum topo I antibody titer between groups treated with or without BTK inhibitors (*Figure 7B*). Furthermore, topo I-APC+ topo I-PE+ CD19+ cells from the topo I-induced SSc model mice treated with the BTK inhibitor exhibited significantly lower affinity for topo I relative to the model mice without the BTK inhibitor (p<0.005; *Figure 7B*). Topo I-APC+ topo I-PE+ CD19+ cells in the topo I-induced SSc model mice treated with the BTK inhibitor had higher frequencies of IL-10 and IL-35-producing cells and also produced the significantly higher amount of IL-10 and IL-35 (p<0.005; *Figure 7C and D*). By contrast, frequencies of topo I-APC+ topo I-PE+ CD19+ cells producing IL-6 and IL-23 were lower in the BTK inhibitor-treated model mice with significantly lower protein expression of IL-6 and IL-23 (p<0.005; *Figure 7C and D*). In addition, adoptive transfer of topo I-APC+ topo I-PE+ CD19+ cells from the topo I-induced SSc model mice treated with the BTK inhibitor failed to induce skin and lung fibrosis in non-immunized WT mice (*Figure 7E*). Thus, BTK inhibition resulted

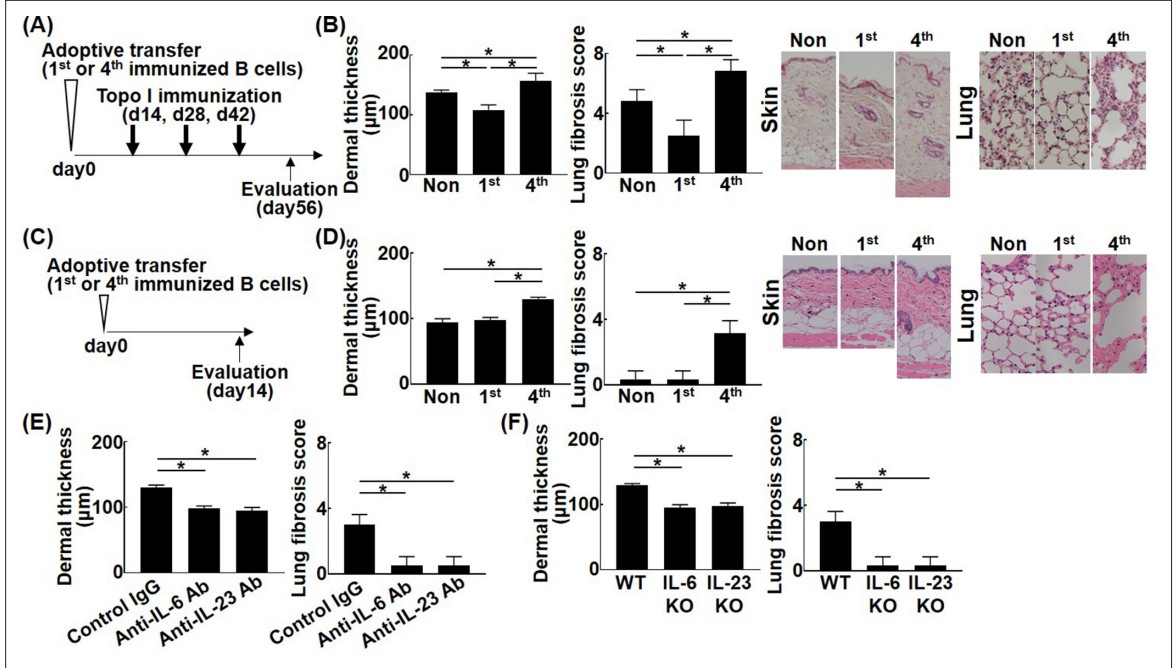

**Figure 6.** Effect of the affinity for topo I in topo I-reactive B cells on the development of fibrosis in the topo I-induced systemic sclerosis (SSc) model mice. Topo I-reactive B cells ($10^4$ cells) from mice immunized once or four times with topo I were adoptively transferred into wild-type (WT) mice, and 14 days later, these WT mice were immunized with topo I three times every 2 weeks (**A**). 56 days after the adoptive transfer, the skin (original magnification, ×40) and lung tissues (×100) were obtained, and the dermal thickness and lung fibrosis score were examined histologically (**B**). Similarly, after 14 days of the adoptive transfer of topo I-reactive B cells ($10^4$ cells) from mice immunized once or four times with topo I into non-immunized WT mice (**C**), the dermal thickness and lung fibrosis score were measured (**D**). Serum anti-topo I antibody levels were elevated in both first and fourth models compared with non-immunized WT mice. Furthermore, topo I-reactive B cells were adoptively transferred with either anti-IL-6 (1 mg/week, administrated subcutaneously) or anti-IL-23 (100 μg/week, administrated subcutaneously) antibodies (Abs), and dermal thickness and lung fibrosis score were examined 14 days after the adoptive transfer (**E**). WT mice, IL-6-deficient (IL-6KO), and IL-23-deficient (IL-23KO) mice were immunized four times with topo I. Then, $10^4$ cells of topo I-reactive B cells obtained were transferred to non-immunized WT mice, and the dermal thickness and lung fibrosis score were measured 14 days later. Serum anti-topo I antibody levels were elevated in both IL-6 KO mice and IL-23 KO mice compared with non-immunized WT mice (**F**). These results represented six experiments. The bar graphs show the mean + SD. *p<0.05.

The online version of this article includes the following figure supplement(s) for figure 6:

**Source data 1.** Source file for the effect of the affinity for topo I in topo I-reactive B cells on the development of fibrosis in the topo I-induced systemic sclerosis (SSc) model mice.

**Figure supplement 1.** The effect of fibrosis and cytokine production by topo I-reactive B cells in the topo I-induced incomplete systemic sclerosis (SSc) model mice.

**Figure supplement 1—source data 1.** Source file for the effect of fibrosis and cytokine production by topo I-reactive B cells in the topo I-induced incomplete systemic sclerosis (SSc) model mice.

**Figure supplement 2.** Effect of B cell-specific Blimp1 or class II in topo I-reactive B cells on the development of fibrosis in the topo I-induced systemic sclerosis (SSc) model mice.

**Figure supplement 2—source data 1.** Source file for the effect of B cell-specific Blimp1 or class II in topo I-reactive B cells on the development of fibrosis in the topo I-induced systemic sclerosis (SSc) model mice.

in the decreased number and affinity for topo I of topo I-reactive B cells, which was associated with increased production of anti-inflammatory cytokines and attenuated fibrosis. For the 10 SSc patients who received RTX treatment, data at 1 year after RTX showed that mRSS improved by an average of about six points, while anti-topo I antibody titer remained the same (*Figure 7—figure supplement 1*). These results suggest that cytokines produced by B cells, rather than antibody production from B cells, are important in the mechanism of action of BTK inhibitors in the pathogenesis of SSc, such as skin sclerosis and ILD.

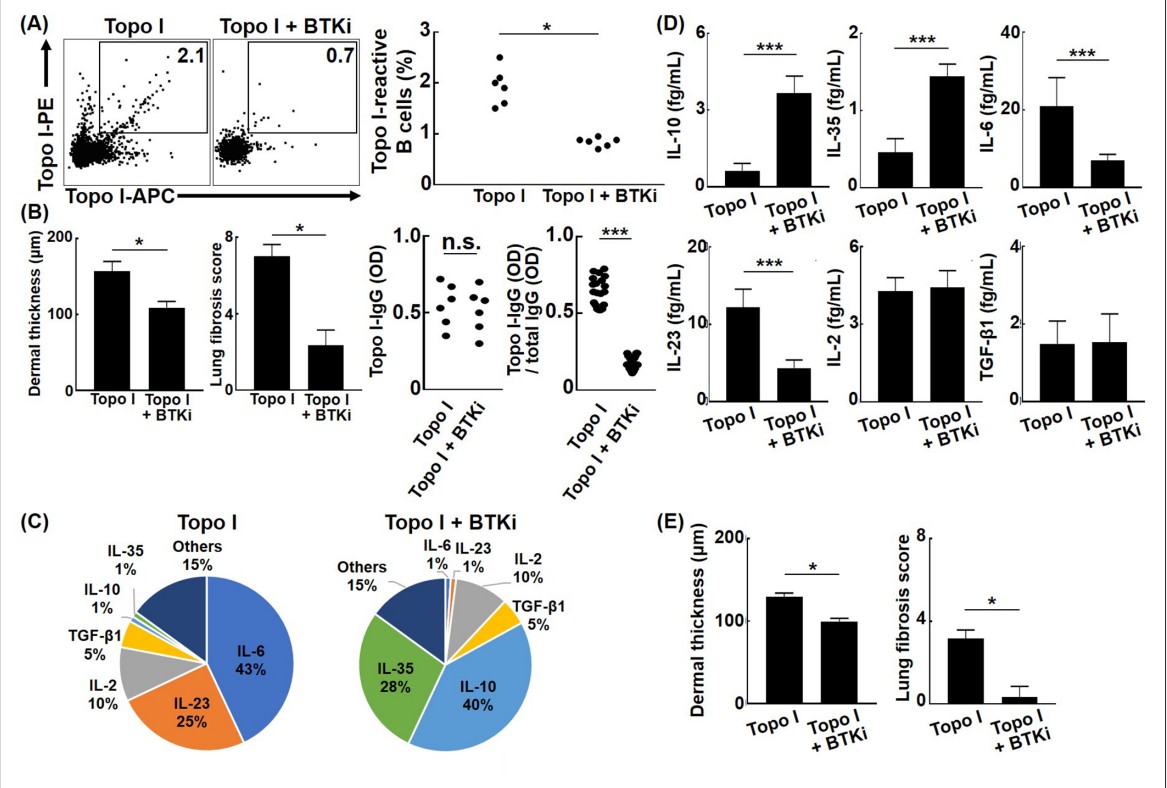

**Figure 7.** Effect of Bruton's tyrosine kinase (BTK) inhibition on the affinity for topo I of topo I-reactive B cells and the development of fibrosis in the topo I-induced systemic sclerosis (SSc) model mice. (**A**) Frequencies of topo I-reactive B cells in splenic B cells from mice immunized four times with topo I and from mice treated with a BTK inhibitor along with topo I were examined. Ibrutinib (12.5 mg/kg/day, administrated orally) was used as the BTK inhibitor. (**B**) We measured the dermal thickness, lung fibrosis score, titer of serum IgG anti-topo I antibodies, and titer of IgG anti-topo I antibodies produced by individual topo I-reactive B cells in these mice (n = 6). Frequencies of cytokine-producing topo I-reactive B cells (**C**, total 200 cells) and the amount of produced cytokine (**D**, 100 cells each) are shown in the pie charts. (**E**) Topo I-reactive B cells ($10^4$ cells) obtained from these mice (n = 6) were adoptively transferred to wild-type mice, and the dermal thickness and lung fibrosis score were measured after 14 days. Topo I, topo I-induced SSc model mice; Topo I + BTKi; BTK inhibitor-treated topo I-induced SSc model mice. The bar graphs show the mean + SD. *p<0.05, ***p<0.005.

The online version of this article includes the following source data and figure supplement(s) for figure 7:

**Source data 1.** Source file for the effect of Bruton's tyrosine kinase (BTK) inhibition on the affinity for topo I of topo I-reactive B cells and the development of fibrosis in the topo I-induced systemic sclerosis (SSc) model mice.

**Source data 2.** Source data for *Figure 7B*.

**Figure supplement 1.** Change of modified Rodnan skin score (mRSS) and anti-topo I antibody from baseline after rituximab (RTX) treatment.

**Figure supplement 1—source data 1.** Source file for change of modified Rodnan skin score (mRSS) and anti-topo I antibody from baseline after rituximab treatment.

## Discussion

The present study shows that topo I-reactive B cells produced a variety of cytokines, the type of which was defined by the degree of affinity to topo I: B cells with low affinity for topo I predominantly produced inhibitory cytokines, while those with high affinity mainly produced pro-inflammatory cytokines. Each cytokine was associated with the corresponding transcription factors or signaling pathways (*Figures 1 and 2*). In addition, topo I-reactive B cells affected Th cell differentiation when they interacted with CD4[+] T cells in the microspace (*Figure 3*): B cells with low-affinity BCRs for topo I differentiated Th cells into regulatory T cells, while B cells with high affinity induced Th17 cell differentiation. Similarly, in the SSc mouse model, B cells with low affinity to topo I inhibited fibrosis, while B cells with high affinity induced or exacerbated it (*Figure 4*). Furthermore, higher frequencies of B cells with high affinity correlated with the more severe fibrosis in SSc patients (*Figure 5* and *Table 1*). Finally, inhibition of pro-inflammatory cytokines produced by topo I-reactive B cells and suppression of increased affinity of BCRs for topo I inhibited fibrosis (*Figures 6 and 7*). Taken together, these results

indicate that autoreactive memory B cells are involved in the development of SSc through their cytokine production and interaction with Th cells.

Autoantibodies in SSc appear early in the onset, and their specificities are closely associated with the disease subsets (*Yoshizaki and Sato, 2015*; *Kayser and Fritzler, 2015*). Since patients with anti-topo I antibody-positive SSc are associated with severe skin sclerosis and interstitial lung disease, the presence of anti-topo I antibodies is a clinical indicator of poor prognosis (*Kayser and Fritzler, 2015*). Although anti-topo I antibody is generally thought to be epiphenomenon (*Sato et al., 2004b*), anti-topo I antibody titers correlate with the activity and severity of SSc (*Sato et al., 2001*), suggesting that topo I-reactive B cells themselves may contribute to the disease manifestations in SSc, independent of anti-topo I antibodies. Consistently, our results show that the increased affinity for topo I of topo I-reactive B cells changed the cytokine production by these B cells from inhibitory IL-10 and IL-35 to pro-inflammatory IL-6 and IL-23 (*Figures 1, 2 and 4*), which was associated with the development of fibrosis (*Figure 5* and *Table 1*). B cells exposed to exogenous antigens, such as viral antigens, increase the affinity of BCR by somatic hypermutation through cognate interaction with T cells (*Hwang* et al., 2015*). The increased affinity of BCR enhances the ability of antibodies to bind to antigens, resulting in efficient elimination of antigens. However, the pathogenic relevance of the increased affinity of antigen-reactive B cells in autoimmune diseases has not been clarified. The results of this study suggest that topo I-reactive B cells with enhanced antigen affinity after long-term exposure to topo I contribute to the development of fibrosis in SSc.

In autoimmune diseases, cytokines are important factors that shape the disease. Indeed, in rheumatoid arthritis, systemic lupus erythematosus, Crohn's disease, and ulcerative colitis, anticytokine therapy has shown excellent therapeutic efficacy and is attracting attention as a new treatment method (*Randall, 2016*). In SSc, a large clinical trial using tocilizumab, an anti-IL-6 receptor antibody, was recently conducted and showed a significant inhibitory effect on the progression of lung fibrosis (*Khanna et al., 2016*). IL-6 enhances collagen production by fibroblasts and directly induces fibrosis (*Dufour et al., 2018*). Various cells, including macrophages, produce IL-6, but it has been suggested that IL-6 produced by B cells is particularly important in SSc (*Yoshizaki, 2016*). However, because cytokines produced by B cells, including IL-6, are trace, as shown in this study (*Figures 2 and 4*), the mechanisms by which cytokine-producing B cells are involved in the development of SSc have not been clarified. In the present study, topo I-specific B cells differentiate CD4+ T cells into regulatory T cells or Th17 cells only when they interact with T cells in the microspace (*Figure 3*). In vivo, cognate interaction of B and T cells takes place primarily in spatially restricted lymphoid follicles. The microspace device used in this study is thought to mimic the B cell-T cell interaction space in vivo, suggesting that cytokines produced by topo I-reactive B cells directly affect T cell differentiation.

Multiple subsets of B cells that produce IL-10, including CD24$^{hi}$ CD27+ B cells, a memory B cell phenotype, have been reported and are collectively referred to as regulatory B cells (*Mauri and Menon, 2017*). As there is no specific phenotype signature for regulatory B cells, they should be regarded by their suppressive function in the immune system rather than by lineage or process of differentiation (*Wąsik et al., 2018*). The current study is the first to identify autoreactive memory B cells with low affinity for topo I that produce IL-10 in human SSc (Figures 2 and 3), which may be regarded as regulatory autoreactive B cells. In the SSc model mice, adoptive transfer of these regulatory autoreactive B cells attenuated fibrosis (*Figure 6*), indicating that regulatory autoreactive B cells have inhibitory functions and may form one of the defense mechanisms that suppress the progression of fibrosis at an early stage. However, upon further exposure to topo I, these cells may transform into pathogenic autoreactive B cells with high affinity for topo I, resulting in the development of fibrosis (*Figure 6*). Recently, B cell depletion therapy has been shown to be effective for SSc (*Yoshizaki, 2016*; *Yoshizaki and Sato, 2015*). Its effectiveness varies from patient to patient (*Mohammed et al., 2017*), suggesting that the therapeutic effect may be determined by relative abundance of pathogenic B cells with high affinity for topo I. Our results also demonstrate that inhibition of affinity maturation to topo I by BTK inhibitors induced regulatory autoreactive B cells and thereby suppressed skin and lung fibrosis (*Figure 7*), suggesting that BTK would be a potential therapeutic target for SSc. On the other hand, cytokine elimination therapy targeting IL-6 and IL-23 produced by pathogenic B cells with high affinity for topo I could also be new treatments for SSc (*Figure 6*). High-affinity topo I antibody-producing B cells produce pro-inflammatory cytokines such as IL-6 and IL-23, while low-affinity topo I antibody-producing B cells produce inhibitory cytokines such as IL-10 and IL-35. In IL-10 producing

B cells, proteins that inhibit signaling pathway, NF-kB inhibitory proteins, MKNK2, p38 kinase, Jak1, stat3, and ATF1 transcription factor were detected, suggesting that IL-10 production is mediated by these proteins. Similarly, regulatory B cells producing IL-35 were found to be mediated by EOLA1, which inhibits IL-6, HPK1, which terminates signaling cascade, and FOS transcription factors. In each of the IL-6- or IL-23-producing effector B cells, the signaling-enhancing proteins were abundant and were associated with distinct signaling factors and transcription factors. These factors also might be a potential therapeutic target for SSc.

The unique single-cell protein analysis used in this study has revealed for the first time that topo I-reactive B cells promote fibrosis as their affinity for topo I increases, suggesting that autoreactive memory B cells contribute to the disease manifestations in SSc. The strategy used in this study can be applied to other autoimmune diseases, which may pave the way for the elucidation of the mechanisms of other autoimmune diseases and the development of new treatments.

## Materials and methods

### Patients

All samples were obtained from each of the 111 anti-topo I antibody-positive SSc patients, 50 anti-CENP antibody-positive SSc patients, and 50 healthy controls, all of whom provided their signed informed consent for the study. All patients fulfilled the American College of Rheumatology classification criteria for SSc (*Alfonse, 1980*). 14 anti-topo I antibody-positive SSc patients had limited cutaneous SSc and 97 patients had diffuse cutaneous SSc according to the disease classification system proposed by *LeRoy et al., 1988*. None of the SSc patients were being treated with oral corticosteroids, D-penicillamine, or any other immunosuppressive therapy at the time of evaluation. All studies were approved by the Committee on Ethics of the University of Tokyo Graduate School of Medicine.

### Clinical assessment of patients

A complete review of the medical history, physical examinations, and laboratory tests, including pulmonary function test, was conducted for all patients. The severity of skin fibrosis was rated using the mRTSS scale of 0–3, where 0 = normal, 1 = mild thickening, 2 = moderate thickening, and 3 = severe thickening, with a maximum possible score of 51, as previously described (*Clements et al., 1993*). Organ involvement was defined in a manner as previously described (*Sato et al., 1994*): for the lungs, interstitial lung disease is defined as bibasilar fibrosis on chest radiography and high-resolution computed tomography.

### The mouse model of topo I-induced SSc

WT C57BL/6 mice, IL-6KO, IL-23KO, CD19-Cre, PRDM1 fl/fl, or MHC class II fl/fl mice were purchased from The Jackson Laboratory (Bar Harbor, ME) or Taconic Biosciences (Rensselaer, NY). These mice were backcrossed between 10 generations onto the C57BL/6 genetic background. All mice were housed in a specific pathogen-free barrier facility and screened regularly for pathogens. The mice used in these experiments were 6 weeks of age. Recombinant topo I proteins were purified from *Escherichia coli* expressing human topo I and were dissolved in saline (100 µg/ml). The topo I solution was mixed 1:1 (volume/volume) with TiterMax Gold adjuvant (Sigma-Aldrich, St. Louis, MO). These solutions (100 µl) were injected up to four times subcutaneously into a single location on the shaved back of the mice with a 26-gauge needle at an interval of 2 weeks, as described previously (*Yoshizaki et al., 2011*; *Fukasawa et al., 2017*). In the mouse studies, at least six mice per group were examined. All studies and procedures were approved by the Committee on Animal Experimentation of the University of Tokyo Graduate School of Medicine.

### Reagents for the treatment of mice

Rat anti-mouse IL-6 monoclonal antibodies (MP5-20F3) were purchased from eBioscience (San Diego, CA). Purified rat IgG1 (Cappel MP Biomedicals, Solon, OH) was administered as an isotype-matched control antibody for the anti-IL-6 antibody. Ultra-LEAF purified anti-IL-23p19 monoclonal antibodies (MMp19B2) and mouse IgG2bκ (MPC-11, control for anti-IL-23p19 antibodies) were purchased from BioLegend (San Diego, CA). Ibrutinib (PCI-32765, 12.5 mg/kg, ab254447, abcam, MA), which was used as a BTK inhibitor, were administered orally once per day (*Figure 7*).

## Purification and stimulation of immune cells

Heparinized blood samples were obtained from SSc patients and healthy controls. In addition, cell suspensions were obtained from the spleen of topo I-induced SSc and control mice. T cells and B cells were collected with an AutoMACS isolation kit (Miltenyi Biotec, Bergisch Gladbach, Germany), respectively. A total of 99% of these cells were CD19+ or CD4+ cells. Antibodies used in this study included fluorescein isothiocyanate (FITC)-conjugated anti-human or anti-mouse CD3 and CD19 antibodies, and phycoerythrin (PE)-Cy7-conjugated anti-human CD27 antibodies (all from BioLegend). In addition, topo I-reactive CD27+ CD19+ cells were determined using PE- or allophycocyanin (APC)-conjugated topo I protein, which were generated using PE or APC labeling kits (AnaSpec, San Jose, CA). Each of the topo I-reactive and topo I-non-reactive CD27+ CD19+ cells and CD4+ cells were purified using a FACSAria flow cytometer (BD Bioscience, San Jose, CA), yielding cell purities of 95–99%. To evaluate the effect of stimulation, ionomycin (500 ng/ml; Sigma-Aldrich), phorbol myristate acetate (PMA) (50 ng/ml; Sigma-Aldrich), and monensin (2 mM; eBioscience) were used for 5 hr (*Figure 1D*). Alexa Fluor 700-conjugated anti-human or mouse CD38 (BioLegend) and VB605-conjugated anti-human or mouse CD95 (BioLegend) staining were performed to determine cell surface markers.

## Workflow of the experiments

Each single cell was sorted to 96-well cell culture plates and analyzed by real-time RT-PCR with or without stimulation (*Figure 1C and D*). To measure cytokines or the affinity of antibodies from single B cells, cells were cultured for 48 hr in 96-well cell culture plates. Cytokines (*Figure 2C and D*) or the affinity of antibodies (*Figures 1B, 2A, C, D, 3A and C*) were analyzed by µELISA system. Each cell was lysed and intracellular proteins were analyzed by µELISA system. Topo I-reactive CD27+ B cells with determined low affinity and high affinity for topo I (100 cells each) and control topo I-non-reactive CD27+ B cells (100 cells) were moved to 96-well cell culture plates or microculture plates and were co-cultured for 48 hr with 10^4 CD4+ T cells with or without blocking antibodies. Subsequent assays (real-time RT-PCR [*Figure 3A and C*] or immunofluorescence cell staining [*Figure 3D and E*]) were conducted.

## ELISA and µELISA system for measurement of cytokine, IgG secretion, and intracellular proteins from single B cells

Serum IgG concentrations were assessed as described previously (*Iwai et al., 2002*). Specific ELISA kits were used to measure anti-topo I and anti-CENP antibodies (Medical & Biological Laboratories, Nagano, Japan). Recently, we successfully integrated an immunoassay into a microchip, applying basic ELISA methods to a µELISA system (*Fukasawa et al., 2017*; *Ohashi et al., 2009*; *Sato et al., 2000*; *Sato et al., 2002*). This system can measure the very low concentration (1–1000 fg/ml) of several proteins using bead-bed immunoassay, which is proceeded on microchip composed by 5–100 µm depth of flow channels. We have also developed a laser-induced thermal lens microscope (TLM), which is especially useful for ultrasensitive determination in a microscope (*Ashcroft et al., 1988*). In this study, the microchip was filled with antibodies-coated polystyrene beads. After beads introduction, 1 µl of samples was loaded into the microchip. In this assay, a highly sensitive detection method with high space resolution is indispensable because there are very small amounts of analytes in a microchip. During the µELISA, dye molecules produced by the enzymatic reaction are detected by our sensitive TLM. A comparison with conventional methods demonstrated excellent performance in sensitivity (1–1000 fg/ml), analysis time (10–20 min), and sample volume (1 µl). To analyze intracellular proteins, 10× RIPA buffer (ab156034, abcam) were used according to the manufacturer's protocol. In this study, the microchip was filled with mouse or human anti-IL-2 (88-7024, 88-7025), IL-6 (88-7064, 88-7066), IL-10 (88-7105, 88-7106), IL-23 (88-7230, 88-7237), human IL-35 (88-7357), TGF-β1 (88-8350) (all from eBioscience), CSK (orb562787), JKAMP (orb548836), JAK2 (orb563817), JAK3 (orb562284), STAT1 (orb564544), STAT5A (orb562049), STAT5B (orb563841) (all from biorbyt, St Louis, MO), MKNK2 (EH10121), ASCC1 (EH6490) (all from FineTest, Wuhan, China), BEX1 (abx503699), HPK1 (abx388409), BRAF (abx150833), C12orf66 (abx507055), CRTC3 (abx151189), JAK1 (abx251876), TYK2 (abx153417), p-STAT2 (pY689, abx153172), STAT3 (abx153173), STAT4 (abx156860), BLK (abx150814), FGR (abx151604) (all from abbexa, Houston, TX), mouse IL-35 (MBS2506294), TNIP2 (MBS2103450), ATF1 (MBS2089082), EOLA1 (MBS7004817), Themis2 (MBS648275), RAF1 (MBS2101444) (all from MyBioSource, San Diego, CA), MAPK14 (LS-F34788-1), FYN (LS-F7527-1) (all from LSBio, Seattle, WA), FOS (ab264626),

NFkB2 (ab207219), CD80 (ab256392), CD86 (ab242239), ICOSL (ab272475), CD40 (ab99990) (all from abcam), HCST (ABIN6212216), p-JAK1 (pY1034/1035, ABIN6255762), LCK (ABIN6229045) (all from antibodies-online, Aachen, Germany), p-JAK2 (pY1007/1008, PEL-JAK2-Y1007-1), p-JAK3 (PEL-JAK3-Y-5), p-TYK2 (PEL-TYK2-Y-2), STAT2 (PEL-STAT2-Y689-T-1), p-STAT4 (pY693, CBEL-STAT4-1), p-STAT5A (pY694, PEL-STAT5A-Y694-T-2), p-BTK (pY551, PEL-BTK-Y551-T-1), LYN (ELH-LYN-5), p-BLK (PEL-BLK-Y-1) (all from RayBiotech, Peachtree Corners, GA), p-STAT1 (pY701, 40716C), STAT6 (7267), p-STAT6 (pY641, 7275), BTK (36609), p-LCK (pY505, 7941) (all from Cell Signaling, Danvers, MA), p-STAT3 (pY705, DYC4607B-2) (R&D Systems, Minneapolis, MN), p-STAT5B (pY699, 85-86112-11) (Thermo Fisher, Waltham, MA), p-LYN (pY507, EKC1990), p-FYN (pY530, EKC2034) (all from Boster Bio, Pleasanton, CA), and IgG antibodies and topo I protein-coated polystyrene beads.

## Single-cell culture and co-culture of topo I-reactive B cells with T cells

Single B cells with T cells were cultured for 48 hr in 96-well cell culture plates at 37°C in a humid atmosphere of 5% $CO_2$ in RPMI 1640 medium (Sigma) freshly supplemented with 10% fetal bovine serum (Hyclone, South Logan, UT) and 1% penicillin/streptomycin (Sigma). Cell culture supernatants from each well were used in subsequent assays. Similarly, these cells were cultured using the microculture plates (Institute of Microchemical Technology, Kanagawa, Japan). The microculture plate is composed of Pyrex glass plates (40 mm × 40 mm) and has 88 microchambers for cell culture and, that is, top and bottom of the microchamber with size of 1500 and 600 μm, respectively, with depth of 700 μm. As reagents, anti-IL-6 (MAB2061), anti-IL-23p19 (AF1716), anti-IL-10 (MAB217), and anti-IL-35 (27537) antibody were purchased from R&D Systems (*Figure 3*).

## Real-time RT-PCR

Each cell, separated to each well in 96-well plates or microculture plates (Institute of Microchemical Technology), was analyzed by Cells-to-CT kits (Thermo Fisher) according to the manufacturer's protocol. Real-time RT-PCR was carried out with a Applied Biosystems StepOnePlus Real-Time PCR System (Thermo Fisher) in a 25 μl reaction volume using SYBR Premix Ex Taq II (Takara Bio) for the detection of PCR products. It was subjected to the recommended RT-PCR thermal cycles: 55°C for 15 min, 94°C for 2 min, 40 cycles of denature (94°C, 15 s), anneal (55°C, 30 s) and extend (68°C, 60 s) steps, and a final extension (68°C, 5 min). Primers for *IL2*, *IL6*, *IL10*, *IL14*, *IL16*, *IL23*, *IL35*, and *TGFB1* were purchased from Thermo Fisher. Primer sequences for *FOXP3* and *RORC* were as follows: *FOXP3* forward, 5'-TCATCCGCTGGGCCATCCTG-3', and *FOXP3* reverse, 5'-GTGGAAACCTCACTTC TTGGTC-3'; *RORC* forward, 5'-TGGACCACCCCCTGCTGAGAAGG-3', and *RORC* reverse, 5'-CTTC AATTTGTGTTCTCATGACT-3'. As control, mRNA content for *GAPDH* was analyzed using the following primers: *GAPDH* forward, 5'-GTGAAGGTCGGAGTCAACG-3', and *GAPDH* reverse, 5'-CAATGCCAG-CCCCAGCG-3'. The expression levels of the *FOXP3* and *RORC* genes were evaluated as the ratio of its mRNA to those of *GAPDH* mRNA.

## Immunofluorescence cell staining

Alexa405-, Alexa488-, and Alexa555-conjugated anti-CD3 (FAB100V, R&D systems), CD20 (2H7, BioLegend), Foxp3 (ab22510, abcam), and RORγt (ab221359, abcam) antibodies were used in this assay, respectively. The cells were fixed with 4% paraformaldehyde for 10 min at room temperature. Cells were blocked with 10% horse serum and incubated with primary antibodies at 37°C for 1 hr and with secondary antibodies at 37°C for 30 min. Images were obtained with an Olympus BX51 fluorescence microscope at ×1000 magnification, captured with Olympus DP 70 digital camera using Olympus DP controller and DP manager software. Signal intensities of Foxp3 and RORγt were analyzed by ImageJ.

## Histopathological assessment of dermal thickness and lung fibrosis

Skin and lung sections obtained from mice were assessed under a light microscope. Sections were stained with hematoxylin and eosin. We examined dermal thickness, which was defined as the thickness of the skin from the top of the granular layer to the junction between the dermis and subcutaneous fat. The severity of lung fibrosis was semiquantitatively assessed according to the method of *Ashcroft et al., 1988*. Briefly, lung fibrosis was graded on a scale of 0–8 by examining randomly chosen fields of the left middle lobe at a magnification of ×100. The grading criteria were as follows:

grade 0, normal lung; grade 1, minimal fibrous thickening of alveolar or bronchiolar walls; grade 3, moderate thickening of walls without obvious damage to lung architecture; grade 5, increased fibrosis with definite damage to lung structure and formation of fibrous bands or small fibrous masses; grade 7, severe distortion of structure and large fibrous areas; and grade 8, total fibrous obliteration of fields. Grades 2, 4, and 6 were used as intermediate pictures between the aforementioned criteria. All of the sections were scored independently by two investigators in a blinded manner.

## Adoptive transfer experiments

Topo I-reactive B cells (topo I-APC$^+$ topo I-PE$^+$ CD19+ cells) from mice immunized once or four times with topo I were purified by cell sorting with purities of 95–98%. After purification, cells were immediately transferred intravenously ($10^4$ cells) to recipient mice. 14 days after adoptive transfer, we analyzed the dermal thickness and severity of lung fibrosis. In some experiments, mice, to which topo I-reactive B cells were adoptively transferred, were immunized with topo I antigens for three times every other week and analyzed histopathologically. Furthermore, topo I-reactive B cells were adoptively transferred with either anti-IL-6 (1 mg/week, administrated subcutaneously) or anti-IL-23 (100 μg/week, administrated subcutaneously) antibodies (Abs), and dermal thickness and lung fibrosis score were examined 14 days after the adoptive transfer. In some experiments, WT mice, IL-6-deficient (IL-6KO) and IL-23-deficient (IL-23KO) mice were immunized four times with topo I. Then, $10^4$ cells of topo I-reactive B cells obtained were transferred to non-immunized WT mice, and the dermal thickness and lung fibrosis score were measured 14 days later.

## Statistical analysis

Statistical analyses were performed using Mann-Whitney $U$-test, the Holm–Bonferroni correction, Steel–Dwass test, Wilcoxon matched-pairs signed-rank test, two-way ANOVA, or Spearman's rank correlation test. The statistical significance was accepted as p-value<0.05. Statistical analyses were performed using JMP (version 13; SAS Institute Inc, Cary, NC).

# Additional information

### Funding
No external funding was received for this work.

### Author contributions
Takemichi Fukasawa, Conceptualization, Data curation, Formal analysis, Investigation, Methodology, Resources, Software, Validation, Visualization, Writing – original draft, Writing – review and editing; Ayumi Yoshizaki, Conceptualization, Data curation, Formal analysis, Investigation, Methodology, Project administration, Resources, Software, Supervision, Validation, Visualization, Writing – original draft, Writing – review and editing; Satoshi Ebata, Asako Yoshizaki-Ogawa, Yoshihide Asano, Atsushi Enomoto, Kiyoshi Miyagawa, Data curation, Formal analysis, Investigation, Methodology, Resources, Software, Validation, Visualization; Yutaka Kazoe, Kazuma Mawatari, Takehiko Kitamori, Data curation, Formal analysis, Investigation, Methodology, Resources, Software, Supervision, Validation, Visualization; Shinichi Sato, Conceptualization, Data curation, Formal analysis, Funding acquisition, Investigation, Methodology, Project administration, Resources, Software, Supervision, Validation, Visualization, Writing – original draft, Writing – review and editing

### Author ORCIDs
Takemichi Fukasawa (iD) http://orcid.org/0000-0002-6093-1881
Ayumi Yoshizaki (iD) http://orcid.org/0000-0002-8194-9140
Shinichi Sato (iD) http://orcid.org/0000-0001-5519-172X

### Ethics
Human subjects: all patients provided their signed informed consent for the study. All studies were approved by the Committee on Ethics of the University of Tokyo Graduate School of Medicine.
This study was performed in strict accordance with the recommendations in the Guide for the Care and Use of Laboratory Animals of the National Institutes of Health. All of the animals were handled

according to approved institutional animal care and use committee (IACUC) protocols (#08-133) of the University of Arizona. The protocol was approved by the Committee on the Ethics of Animal Experiments of the University of Tokyo (Permit Number: P18-089). All surgery was performed under sodium pentobarbital anesthesia, and every effort was made to minimize suffering.

### Decision letter and Author response
Decision letter https://doi.org/10.7554/eLife.67209.sa1
Author response https://doi.org/10.7554/eLife.67209.sa2

## Additional files

### Supplementary files
• Transparent reporting form

### Data availability
All data generated or analysed during this study are included in the manuscript and supporting files.

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
