## [Editor Report]

By using human samples and the recaptured mouse model, the authors nicely showed the importance of B cell-mediated cytokines in autoimmune diseases.

---

## [Decision Letter]

**Decision letter after peer review:**

Thank you for submitting your article "Single cell level protein analysis revealing the roles of autoantigen-reactive B lymphocytes in autoimmune disease" for consideration by *eLife*. Your article has been reviewed by 3 peer reviewers, one of whom is a member of our Board of Reviewing Editors, and the evaluation has been overseen by Betty Diamond as the Senior Editor. The reviewers have opted to remain anonymous.

Essential revisions:

1) The authors claimed that B cells with "high affinity" B cell receptor (BCR) to topo I induce differentiation of Th cells and fibrosis whereas B cells with "low affinity" BCR play protective roles. However, their definition of high and low "affinity" throughout this paper is problematic. They defined "high affinity group" as topo I-reactive B cells that secrete high titer anti-topo I IgG divided by total IgG. But what they measured with microfluidic ELISA (μELISA) was "titer" of IgG that was secreted from in vitro cultured B cells. Affinity of antibody was not directly measured in this study. Because the conclusions throughout the manuscript refer back to this analysis, it would be important to take an orthogonal approach to measure affinity (and/or SHM) directly in human samples.

2) The authors sorted CD27+CD19^+^ B cells with topo I reactive BCR from SSc patients. These are referred to as memory B cells, but this fraction includes memory B cells and also plasmablasts (https://pubmed.ncbi.nlm.nih.gov/31681331/). Memory B cells are usually in resting state and don't produce antibody efficiently unless stimulated to differentiate into plasma cells. Based on IgG production ex vivo, a considerable amount of plasmablasts may be among the sorted cells. And the high proportion of plasmablasts in CD27+CD19^+^ B cells in SSc patients could account for the high titer of IgG relative to healthy controls. Actually, it looks like there are two distinct populations within topo-I reactive cells based on IgG production according to Figure 2A. If "high affinity B cells" are mainly plasmablasts and "low affinity B cells" are memory B cells, the cytokine data are consistent with that secreted from each cell type. For instance, circulating plasmablasts are known to produce IL-6 (J Immunol. 2015;194(6):2482-5.). The authors should address this possibility by staining CD38 and/or CD138 for CD27+CD19^+^ B cells.

Moreover, the correlation between the ratio for high affinity B cells/low affinity B cells and clinical parameters could be explained by accumulation of plasmablasts over time. Skin fibrosis and lung function get worse with disease duration. In fact, "high affinity B cell dominant group" had longer disease duration than "low affinity dominant group". The authors should investigate whether accumulation of "high affinity B cells" (or plasmablasts if proved) correlates with disease duration. If correlated with disease duration, the argument that "high affinity B cells" correlate with disease severity does not make sense.

3) The authors used topo I-induced SSc model mice to investigate the role of "high affinity B cells". They compared topo-I reactive B cells from the mice that immunized only once and immunized 4 times. The B cells in 4 times immunized mice is clearly qualitatively different from prime immunization since they had been boosted and stronger immune reactions were induced. So, I don't think it is a fair comparison and the data seems merely reflection of the differences between prime versus boost responses. It is more compelling to compare the cells with strongly and weakly topo-I reactivity within the same animal. Moreover, related to point 2 above, gated mouse B cells are CD19^+^ Topo I binding. It would be important to clarify whether plasmablasts or memory B cells or even GC B cells are being pulled out and sorted, and whether those phenotypes help account for distinct cytokine profiles.

4) Figure 7. Treatment with a BTK inhibitor might reduce production of anti-TOPOI antibody, and reduction of anti-TOPOI antibody might be involved in modulation of disease severity. The authors should address the contribution of antibody production vs. cytokine production in the beneficial effect of BTK inhibitor.

5) Despite the interesting manuscript, by itself it is not well written. This should be re-written carefully and thoughtfully.

*Reviewer #1:*

At least, three B cells action mechanisms for inducing and/or amplifying autoimmune diseases exist; 1) Ab produced by plasma cells; 2) cytokines produced by B cells; 3)activation of T cells.

By using human patient samples from systemic sclerosis (SSc) and employing mouse model system presumably reflecting human pathology, authors identified high-affinity pathogenic B cells , which produce inflammatory cytokine, being responsible for making pathology at the organism level. Overall quality of this manuscript is high, but some important experiments are missing to dissect the above three potential mechanisms. And, the manuscript by itself is not so well written; authors are encouraged to focus what they really want to appeal.

Despite the existence of some weak points in this manuscript, I think that this is a nice study, providing a significant impact in this area. So, I recommend its publication after some revisions.

Comments for the authors:

By using human patient samples from systemic sclerosis (SSc) and employing mouse model system presumably reflecting human pathology, authors identified high-affinity pathogenic B cells , which produce inflammatory cytokine, being responsible for making pathology at the organism level. Overall quality of this manuscript is high, but some important experiments are missing to make the conclusions more solid.

Moreover, the manuscript by itself is not well written; authors are encouraged to make their conclusions more appealing.

First, complementary experiments for Figure 6 should be carried out. The question is whether high affinity Abs play no role in pathogenicity. To test this idea, authors are encouraged to use Blimp1 ko B cells.

Second, probably B-T interaction is required. So, to test this directly, B cell-specific Class II ko should be used.

Third, assuming that B-T interaction is required, how can high-affinity B cells recognizing autoantigens activate presumably cognate T cells? In this context, authors should at least examine the status of CD80/CD86, ICOSL, CD40.

*Reviewer #2:*

In this manuscript, Fukasawa et al. isolated TopoI-reactive CD19^+^ CD27+ memory B cells from patient with systemic sclerosis (SSc), and demonstrated difference in cytokine production between high and low affinity B cells. High affinity B cells produce inflammatory cytokines whereas low affinity B cells produce inhibitory cytokines. They showed that co-culture of low and high affinity B cells induces differentiation of CD4 T cells to different T helper cell subsets, and that the frequency of high affinity B cells is correlated to fibrotic symptoms in SSc patients, suggesting that cytokine production of low and high affinity B cells regulates disease severity. They also showed similar findings in a mouse model and potential therapeutic effect of a BTK inhibitor. The differential cytokine production in high vs. low affinity B cells and its role in the regulation of disease severity are novel and interesting. However, cytokine production from B cells may be affected by factors other than antibody affinity such as inflammation, and disease severity may be affected by other factors such as the titer of anti-TOPOI antibody. In this study, the authors did not carefully address the role of these other factors. Also, the authors fail to address an important question whether affinity-dependent cytokine production of B cells is restricted to TOPOI-reactive B cells or applicable to antigen-stimulated B cells in general.

Comments for the authors:

1) Figure 1. The authors addressed anti-TopoI IgG production from TOPOI-reactive CD19^+^ CD27+ cells cultured without stimulation. Although the authors describe CD27+ cells as memory B cells in the result section, memory B cells are thought to produce antibodies only after differentiation to plasma blasts. Are these IgG-secreting CD27+ cells memory B cells or plasma blasts? Related to this point, the authors should also show how much fraction of CD27+ cells produce IgG. Do TopoI-reactive B cells from healthy individuals and SSc-CENP patients produce anti-TopoI antibody when they are stimulated?

2) Figure 4. The authors isolated high and low affinity anti-TopoI IgG-producing B cells from mice immunized with TopoI once and four times, respectively, and demonstrated differential cytokine production between high and low affinity B cells. The mice immunized four times may have more severe inflammation than those immunized once. The differential cytokine production might be the consequence of the severity of inflammation. To exclude this possibility, the authors may be able to use low affinity B cells obtained by immunization with TopoI once followed by injection of adjuvant alone three times.

3) Figure 6A-D. The authors transferred TopoI-reactive B cells from mice immunized with TopoI. The authors should show whether sufficient amount of anti-TOPOI IgG is produced by transfer of TopoI-reactive B cells.

4) Figure 6F. The authors transferred TOPO-I-reactive B cells from mice deficient in IL-6 or IL-23. The authors should show whether production of anti-TOPOI antibody in these recipient mice is comparable to those transferred with wild-type B cells. Treatment with anti-IL-6 and anti-IL-23 antibody after transfer (Figure 6E) may circumvent this problem, but these antibodies might modulate non-B cells.

5) Figure 7. Treatment with a BTK inhibitor might reduce production of anti-TOPOI antibody, and reduction of anti-TOPOI antibody might be involved in modulation of disease severity. The authors should address the contribution of antibody production vs. cytokine production in the beneficial effect of BTK inhibitor.

6) It is important to address whether differential cytokine production between low and high affinity B cells is restricted to TOPO-I-reactive B cells. This may be done by immunizing the mice with foreign antigens and examining cytokine production from low and high affinity B cells.

*Reviewer #3:*

The authors sought to argue that in patients and an SSc mouse model, B cells with high or low affinity binding to autoantigen Topo I have distinct pro- and anti-inflammatory cytokine profiles and this can in turn promote or ameliorate disease. Use of both human and mouse approaches, a well-characterized SSc patient cohort, as well as powerful single cell functional technologies are strengths. However, there are major concerns about interpretation of the data and identity of B cell subsets captured in this study:

Authors claim that high affinity and low affinity Topo I-specific B cells are captured and profiled from patients but:

(a) Affinity is never directly measured.

(b) CD27+ gate can capture both memory and plasmablast human B cell populations and this latter distinction could account for differences in Ab measures and cytokine profiles rather than 'affinity'.

Finally, use of mouse model is laudable, but comparison between 1x immunized and 4x immunized CD19^+^ Ag-specific B cells is not fair since the b cell populations may be very different (memory v. plasmablast v. GC). It would be appropriate to characterize B cell subsets and also to make comparison of low and high affinity B cells within individual mice to support claims of the paper.

Comments for the authors:

1. The authors claimed that B cells with "high affinity" B cell receptor (BCR) to topo I induce differentiation of Th cells and fibrosis whereas B cells with "low affinity" BCR play protective roles. However, their definition of high and low "affinity" throughout this paper is problematic. They defined "high affinity group" as topo I-reactive B cells that secrete high titer anti-topo I IgG divided by total IgG. But what they measured with microfluidic ELISA (μELISA) was "titer" of IgG that was secreted from in vitro cultured B cells. Affinity of antibody was not directly measured in this study. Because the conclusions throughout the manuscript refer back to this analysis, it would be important to take an orthogonal approach to measure affinity (and/or SHM) directly in human samples.

2. The authors sorted CD27+CD19^+^ B cells with topo I reactive BCR from SSc patients. These are referred to as memory B cells, but this fraction includes memory B cells and also plasmablasts (https://pubmed.ncbi.nlm.nih.gov/31681331/). Memory B cells are usually in resting state and don't produce antibody efficiently unless stimulated to differentiate into plasma cells. Based on IgG production ex vivo, a considerable amount of plasmablasts may be among the sorted cells. And the high proportion of plasmablasts in CD27+CD19^+^ B cells in SSc patients could account for the high titer of IgG relative to healthy controls. Actually, it looks like there are two distinct populations within topo-I reactive cells based on IgG production according to Figure 2A. If "high affinity B cells" are mainly plasmablasts and "low affinity B cells" are memory B cells, the cytokine data are consistent with that secreted from each cell type. For instance, circulating plasmablasts are known to produce IL-6 (J Immunol. 2015;194(6):2482-5.). The authors should address this possibility by staining CD38 and/or CD138 for CD27+CD19^+^ B cells.

Moreover, the correlation between the ratio for high affinity B cells/low affinity B cells and clinical parameters could be explained by accumulation of plasmablasts over time. Skin fibrosis and lung function get worse with disease duration. In fact, "high affinity B cell dominant group" had longer disease duration than "low affinity dominant group". The authors should investigate whether accumulation of "high affinity B cells" (or plasmablasts if proved) correlates with disease duration. If correlated with disease duration, the argument that "high affinity B cells" correlate with disease severity does not make sense.

3. The authors used topo I-induced SSc model mice to investigate the role of "high affinity B cells". They compared topo-I reactive B cells from the mice that immunized only once and immunized 4 times. The B cells in 4 times immunized mice is clearly qualitatively different from prime immunization since they had been boosted and stronger immune reactions were induced. So, I don't think it is a fair comparison and the data seems merely reflection of the differences between prime versus boost responses. It is more compelling to compare the cells with strongly and weakly topo-I reactivity within the same animal. Moreover, related to point 2 above, gated mouse B cells are CD19^+^ Topo I binding. It would be important to clarify whether plasmablasts or memory B cells or even GC B cells are being pulled out and sorted, and whether those phenotypes help account for distinct cytokine profiles.

[Editors' note: further revisions were suggested prior to acceptance, as described below.]

Thank you for resubmitting your work entitled "Single cell level protein analysis revealing the roles of autoantigen-reactive B lymphocytes in autoimmune disease and the murine model" for further consideration by *eLife*. Your revised article has been evaluated by Betty Diamond as the Senior Editor, and a Reviewing Editor.

The manuscript has been improved but there are some remaining issues that need to be addressed, as outlined below:

In regard to the 4) point among essential revisions, authors have not addressed experimentally the contribution of antibody production, which is important in this manuscript.

---

## [Author Response]

Essential revisions:1) The authors claimed that B cells with "high affinity" B cell receptor (BCR) to topo I induce differentiation of Th cells and fibrosis whereas B cells with "low affinity" BCR play protective roles. However, their definition of high and low "affinity" throughout this paper is problematic. They defined "high affinity group" as topo I-reactive B cells that secrete high titer anti-topo I IgG divided by total IgG. But what they measured with microfluidic ELISA (μELISA) was "titer" of IgG that was secreted from in vitro cultured B cells. Affinity of antibody was not directly measured in this study. Because the conclusions throughout the manuscript refer back to this analysis, it would be important to take an orthogonal approach to measure affinity (and/or SHM) directly in human samples.

Thank you for your question. As you mentioned, we conducted orthogonal affinity analysis in “high affinity” (=high anti-Topo I IgG/total IgG, n = 10) and “low affinity” (=low anti- Topo I IgG/total IgG, n = 10) Topo I-PE+ Topo I-APC+ CD19^+^CD27+ B cells in PBMCs in anti-Topo I antibody positive SSc patients. (Figure 2—figure supplement 1) As a result,”high affinity” B cells defined by titers have low Kd value (= high affinity), and ”low affinity” B cells defined by titers have high Kd value (= low affinity). -log_10_Kd was correlated with Topo I-IgG(OD)/total IgG(OD). The correlation coefficient was r=0.96, and p-value was p<0.0001 and significant. In this analysis system, we can think of titer as affinity. “High affinity” B cells defined by both titers or Kd value produce inflammatory cytokines such as IL-6 or IL-23. “Low affinity” B cells defined by both titers or Kd value also produce regulatory cytokines such as IL-10 or IL-35.

2) The authors sorted CD27+CD19^+^ B cells with topo I reactive BCR from SSc patients. These are referred to as memory B cells, but this fraction includes memory B cells and also plasmablasts (https://pubmed.ncbi.nlm.nih.gov/31681331/). Memory B cells are usually in resting state and don't produce antibody efficiently unless stimulated to differentiate into plasma cells. Based on IgG production ex vivo, a considerable amount of plasmablasts may be among the sorted cells. And the high proportion of plasmablasts in CD27+CD19^+^ B cells in SSc patients could account for the high titer of IgG relative to healthy controls. Actually, it looks like there are two distinct populations within topo-I reactive cells based on IgG production according to Figure 2A. If "high affinity B cells" are mainly plasmablasts and "low affinity B cells" are memory B cells, the cytokine data are consistent with that secreted from each cell type. For instance, circulating plasmablasts are known to produce IL-6 (J Immunol. 2015;194(6):2482-5.). The authors should address this possibility by staining CD38 and/or CD138 for CD27+CD19^+^ B cells.Moreover, the correlation between the ratio for high affinity B cells/low affinity B cells and clinical parameters could be explained by accumulation of plasmablasts over time. Skin fibrosis and lung function get worse with disease duration. In fact, "high affinity B cell dominant group" had longer disease duration than "low affinity dominant group". The authors should investigate whether accumulation of "high affinity B cells" (or plasmablasts if proved) correlates with disease duration. If correlated with disease duration, the argument that "high affinity B cells" correlate with disease severity does not make sense.

Thank you for your questions. We stained Topo I-PE+ Topo I-APC+ CD19^+^CD27+ B cells in PBMCs in anti-Topo I antibody positive SSc patients with anti-CD38 antibody and anti-CD95 antibody. As a result, Topo I reactive fraction was almost all CD95+CD38-. The frequencies of these Topo I reactive fraction were significantly higher than that of TopoI non-reactive fraction (Figure 2—figure supplement 2). Therefore, these Topo I reactive CD19^+^CD27+ fraction was CD38-CD95+, which was activated memory B cells.

The titer of Topo I-non-reactive B cells fraction, or Topo I-reactive B cells fraction in HC or SSc-CENP were not elevated. It was equivalent to background. That means they do not produce anti-Topo I antibody. Almost all of the topo I-reactive B cells in SSc-Topo I produced anti-Topo I antibody. With regard to SSc-Topo I, the Topo I titer divided by total IgG (Topo I-IgG/total IgG) was high or low in B cells that actually produced Topo I antibodies. When we measured the affinity (Kd value) of these samples using the orthogonal approach, they still corresponded to high affinity and low affinity, respectively. We found that both were the same, activated memory B cells, but showed differences in their affinity. Later experiments showed that these differences in affinity corresponded to differences in cytokine production.

From the above experiment, the Topo I reactive fraction was found to be CD38 negative, indicating that it was not a plasmablast but activated memory B cells. As you pointed out, we analyzed the correlation between the duration of the disease and the ratio for high affinity B cells/low affinity B cells, and the correlation coefficient was r=0.69, p<0.0001, which is a significant correlation. The relationship between duration and disease severity is inextricably linked. It is true that the human correlation data suggests that high affinity B cells (not plasmablast) may be responsible for the accumulation, but later mouse experiments have shown that high affinity B cells have inflammatory and pathogenic functions, while low affinity B cells have inhibitory and regulatory functions. In addition, since the patient group in this study was untreated, we believe that our results are consistent with clinical experience in that the longer the duration of the disease (the later the detection), the more severe the indicators such as skin hardening and ILD become.

3) The authors used topo I-induced SSc model mice to investigate the role of "high affinity B cells". They compared topo-I reactive B cells from the mice that immunized only once and immunized 4 times. The B cells in 4 times immunized mice is clearly qualitatively different from prime immunization since they had been boosted and stronger immune reactions were induced. So, I don't think it is a fair comparison and the data seems merely reflection of the differences between prime versus boost responses. It is more compelling to compare the cells with strongly and weakly topo-I reactivity within the same animal. Moreover, related to point 2 above, gated mouse B cells are CD19^+^ Topo I binding. It would be important to clarify whether plasmablasts or memory B cells or even GC B cells are being pulled out and sorted, and whether those phenotypes help account for distinct cytokine profiles.

As you pointed out, we analyzed Topo I-reactive B cells obtained from four immunized mouse models (Figure 4—figure supplement 1) The cytokine profile was similar to human, with more inhibitory cytokines at low affinity and more pro-inflammatory cytokines at high affinity (Figure 4—figure supplement 1B). Cytokine production was also higher at low affinity for inhibitory cytokines and higher at high affinity for pro-inflammatory cytokines (Figure 4—figure supplement 1C). Analysis of Topo I-PE+Topo I-APC+CD19^+^ B cells showed that most of the Topo I reactive B cells were CD38-CD95+ (Figure 4—figure supplement 1D). On the other hand, in Topo I non-reactive B cells, the percentage of CD38-CD95+ was only a few percent (Figure 4—figure supplement 1D). Of the Topo I reactive B cells, both Low affinity B cells and High affinity B cells were obtained from this fraction, suggesting that affinity, not cell surface markers, is what is important in determining cytokine profiles and production. It was found to be CD38- and that means they are not plasmablasts.

A rigorous examination of memory B cells in mouse spleen must be performed several months after TopoI immunization. Due to the short period of time between immunization and collection, it was difficult to rigorously examine resting memory in this model. Based on the results of CD38 and CD95 staining in this study, it was thought that the study was performed with the difference by TopoI reactivity (high vs. low) of activated B cells or B cells in the germinal center.

4) Figure 7. Treatment with a BTK inhibitor might reduce production of anti-TOPOI antibody, and reduction of anti-TOPOI antibody might be involved in modulation of disease severity. The authors should address the contribution of antibody production vs. cytokine production in the beneficial effect of BTK inhibitor.

In SSc, there have been many reports that antibodies themselves are pathogenic or not pathogenic. The results suggest that BTK inhibitors inhibit B cells, especially high affinity B cells, and suppress the production of inflammatory cytokines such as IL-6 and IL-23, which are produced by B cells, results in suppression of skin fibrosis and lung fibrosis.　 It is possible that suppressing antibody production modifies disease severity, but we believe that the effect of suppressing cytokines is more likely.　We believe that this idea is also suggested by the results using anti-cytokine antibodies and cytokine knock out mice.

5) Despite the interesting manuscript, by itself it is not well written. This should be re-written carefully and thoughtfully.

Thank you for pointing this out. I rewrote it based on the above.

Reviewer #1:[…] By using human patient samples from systemic sclerosis (SSc) and employing mouse model system presumably reflecting human pathology, authors identified high-affinity pathogenic B cells , which produce inflammatory cytokine, being responsible for making pathology at the organism level. Overall quality of this manuscript is high, but some important experiments are missing to make the conclusions more solid.Moreover, manuscript by itself is not well written; authors are encouraged to make their conclusions more appealing.First, complementary experiments for Figure 6 should be carried out. The question is whether high affinity Abs play no role in pathogenicity. To test this idea, authors are encouraged to use Blimp1 ko B cells.Second, probably B-T interaction is required. So, to test this directly, B cell-specific Class II ko should be used.Third, assuming that B-T interaction is required, how can high-affinity B cells recognizing autoantigens activate presumably cognate T cells? In this context, authors should at least examine the status of CD80/CD86, ICOSL, CD40.

Thanks for pointing this out. As you mentioned, we generated and analyzed B cell-specific Blimp1 CKO mice and B cell-specific Class II CKO mice (Figure 6—figure supplement 2). Since the transfer of TopoI-reactive B cells from B cell-specific Blimp1 CKO mice induced skin sclerosis and lung fibrosis, it is likely that activated memory B cells play an important role in this model, rather than plasmablast or plasma cells, as we have analyzed (Figure 4—figure supplement 1D). The transfer of Topo I-reactive B cells in B cell-specific Class II CKO mice did not induce skin thickness or lung fibrosis, suggesting that B-T interaction is still important.

In this model, it was thought that B cells activated T cells, causing skin fibrosis and lung fibrosis.　The expression levels of CD80/86, ICOSL, and CD40 in TopoI-PE+TopoI-APC+CD19^+^CD27+ B cells were examined using micro-ELISA, and both were significantly up-regulated compared to Topo I non-reactive B cells. These co-stimulatory molecules were not related to affinity, and the combination of affinity and cytokines was thought to determine the direction of T cell differentiation (Figure 2—figure supplement 3).

Reviewer #2:[…] 1) Figure 1. The authors addressed anti-TopoI IgG production from TOPOI-reactive CD19^+^ CD27+ cells cultured without stimulation. Although the authors describe CD27+ cells as memory B cells in the result section, memory B cells are thought to produce antibodies only after differentiation to plasma blasts. Are these IgG-secreting CD27+ cells memory B cells or plasma blasts? Related to this point, the authors should also show how much fraction of CD27+ cells produce IgG. Do TopoI-reactive B cells from healthy individuals and SSc-CENP patients produce anti-TopoI antibody when they are stimulated?

Thank you for your question. The results of staining with CD38 and CD95 showed that the TOPOI-reactive CD19^+^ CD27+ cells were CD38-CD95+, indicating that they were not plasmablasts, but activated memory B cells. Almost all of these fractions yielded significant antibody titers using micro-ELISA. The fractions of TopoI-responsive B cells obtained from HCs and SSc-CENPs did not produce antibodies, nor did they produce antibodies upon stimulation, suggesting that the reaction was nonspecific. Since we could not obtain any significant titer, we could not analyze the affinity using the orthogonal approach.

2) Figure 4. The authors isolated high and low affinity anti-TopoI IgG-producing B cells from mice immunized with TopoI once and four times, respectively, and demonstrated differential cytokine production between high and low affinity B cells. The mice immunized four times may have more severe inflammation than those immunized once. The differential cytokine production might be the consequence of the severity of inflammation. To exclude this possibility, the authors may be able to use low affinity B cells obtained by immunization with TopoI once followed by injection of adjuvant alone three times.

Thank you for pointing this out. The study was conducted using the TopoI 1 time + Adjuvant 3 time model (Incomplete model Figure 6—figure supplement 1A). Skin thickness and lung fibrosis were comparable to the PBS group (Figure 6—figure supplement 1B). The number of Topo I-reactive B cells was significantly higher than that in the PBS group (Figure 6—figure supplement 1C), and the antibody titer produced by single B cells was also significantly higher (Figure 6—figure supplement 1D). The obtained cytokine profile was similar to that of the single-immunization mouse model (Figure 6—figure supplement 1E). TopoI-reactive B cells in incomplete model had significantly higher levels of inhibitory cytokines such as IL-10 and IL-35 and significantly lower levels of pro-inflammatory cytokines such as IL-6 and IL-23 compared to the complete model (Figure 6—figure supplement 1F). Similar to the single-immunization model mice, skin hardening and lung fibrosis were suppressed in the Incomplete group compared with the non-immunized group in the model immunized with Topo I after adoptive transfer (Figure 6—figure supplement 1G, 1H). Adoptive transfer alone showed no significant change in the Incomplete group compared to the non-implanted group (Figure 6—figure supplement 1I, 1J).

3) Figure 6A-D. The authors transferred TopoI-reactive B cells from mice immunized with TopoI. The authors should show whether sufficient amount of anti-TOPOI IgG is produced by transfer of TopoI-reactive B cells.

Thanks for the question. The antibody titer of Topo I was elevated. We believe the transplant is going well.

4) Figure 6F. The authors transferred TOPO-I-reactive B cells from mice deficient in IL-6 or IL-23. The authors should show whether production of anti-TOPOI antibody in these recipient mice is comparable to those transferred with wild-type B cells. Treatment with anti-IL-6 and anti-IL-23 antibody after transfer (Figure 6E) may circumvent this problem, but these antibodies might modulate non-B cells.

Thank you for your question. The anti-TopoI antibody titer was similarly elevated in IL-6KO, IL-23KO, and WT mice. Therefore, we thought that cytokine production would have a greater impact on skin fibrosis and lung fibrosis than antibody titer. The importance of cytokines is also suggested by the fact that it is known that antibody titer itself does not decrease easily even if symptoms improve after cytokines are suppressed by treatment of scleroderma.

5) Figure 7. Treatment with a BTK inhibitor might reduce production of anti-TOPOI antibody, and reduction of anti-TOPOI antibody might be involved in modulation of disease severity. The authors should address the contribution of antibody production vs. cytokine production in the beneficial effect of BTK inhibitor.

Thank you for your question. From the experiment shown in Figure 6, it was thought that the effect of inhibiting cytokine production on skin fibrosis and lung fibrosis was greater than that of antibody titer. The Topo I-IgG(OD)/total IgG(OD) values in Figure 6B represent those obtained from individual single B cells, and are thought to correspond to affinity from the experiment in Figure 2—figure supplement 1. We believe that the BTK inhibitor reduced the number of high affinity B cells and increased the number of low affinity B cells, resulting in a shift from significant pro-inflammatory cytokines to significant inhibitory cytokines, thereby suppressing skin fibrosis and lung fibrosis. As a result, it is cytokines that are functioning, but we believe that affinity is important in producing cytokines. Upstream is reactivity to self-antigens, and downstream is cytokines. It is thought that the difference in affinity to self-antigens changes the ability to produce cytokines.

6) It is important to address whether differential cytokine production between low and high affinity B cells is restricted to TOPO-I-reactive B cells. This may be done by immunizing the mice with foreign antigens and examining cytokine production from low and high affinity B cells.

We think that this is probably true for other proteins as well. The usefulness of anti-cytokine therapy and B cell removal therapy suggests that it is not the antibody titer but the B cells and cytokines produced by the B cells that are important in the pathogenesis. Since there are not many models to date, we hope to consider this in future experiments.

Reviewer #3:[…] 1. The authors claimed that B cells with "high affinity" B cell receptor (BCR) to topo I induce differentiation of Th cells and fibrosis whereas B cells with "low affinity" BCR play protective roles. However, their definition of high and low "affinity" throughout this paper is problematic. They defined "high affinity group" as topo I-reactive B cells that secrete high titer anti-topo I IgG divided by total IgG. But what they measured with microfluidic ELISA (μELISA) was "titer" of IgG that was secreted from in vitro cultured B cells. Affinity of antibody was not directly measured in this study. Because the conclusions throughout the manuscript refer back to this analysis, it would be important to take an orthogonal approach to measure affinity (and/or SHM) directly in human samples.

Thank you for your question. In the essential revision 1) above, as an orhogonal approach to measure affinity, micro-ELISA was used to analyze affinity in human samples. Since Titer and affinity were correlated, we thought that anti-TopoI IgG/total IgG could be considered as affinity.

2. The authors sorted CD27+CD19^+^ B cells with topo I reactive BCR from SSc patients. These are referred to as memory B cells, but this fraction includes memory B cells and also plasmablasts (https://pubmed.ncbi.nlm.nih.gov/31681331/). Memory B cells are usually in resting state and don't produce antibody efficiently unless stimulated to differentiate into plasma cells. Based on IgG production ex vivo, a considerable amount of plasmablasts may be among the sorted cells. And the high proportion of plasmablasts in CD27+CD19^+^ B cells in SSc patients could account for the high titer of IgG relative to healthy controls. Actually, it looks like there are two distinct populations within topo-I reactive cells based on IgG production according to Figure 2A. If "high affinity B cells" are mainly plasmablasts and "low affinity B cells" are memory B cells, the cytokine data are consistent with that secreted from each cell type. For instance, circulating plasmablasts are known to produce IL-6 (J Immunol. 2015;194(6):2482-5.). The authors should address this possibility by staining CD38 and/or CD138 for CD27+CD19^+^ B cells.Moreover, the correlation between the ratio for high affinity B cells/low affinity B cells and clinical parameters could be explained by accumulation of plasmablasts over time. Skin fibrosis and lung function get worse with disease duration. In fact, "high affinity B cell dominant group" had longer disease duration than "low affinity dominant group". The authors should investigate whether accumulation of "high affinity B cells" (or plasmablasts if proved) correlates with disease duration. If correlated with disease duration, the argument that "high affinity B cells" correlate with disease severity does not make sense.

Thank you for your question. I have answered this question in Essential revision 2) above.

3. The authors used topo I-induced SSc model mice to investigate the role of "high affinity B cells". They compared topo-I reactive B cells from the mice that immunized only once and immunized 4 times. The B cells in 4 times immunized mice is clearly qualitatively different from prime immunization since they had been boosted and stronger immune reactions were induced. So, I don't think it is a fair comparison and the data seems merely reflection of the differences between prime versus boost responses. It is more compelling to compare the cells with strongly and weakly topo-I reactivity within the same animal. Moreover, related to point 2 above, gated mouse B cells are CD19^+^ Topo I binding. It would be important to clarify whether plasmablasts or memory B cells or even GC B cells are being pulled out and sorted, and whether those phenotypes help account for distinct cytokine profiles.

Thanks for your question. I have answered this question in Essential revision 3) above.

[Editors' note: further revisions were suggested prior to acceptance, as described below.]

The manuscript has been improved but there are some remaining issues that need to be addressed, as outlined below:In regard to the 4) point among essential revisions, authors have not addressed experimentally the contribution of antibody production, which is important in this manuscript.

Thank you for your question. In Topo I-induced SSc model mice, there was no difference in serum anti-Topo I level between mice treated with or without BTK inhibitors (newly added Figure 7B). In contrast, skin sclerosis and ILD were improved by BTK inhibitor treatment (Figure 7B). As an explanation for the discrepancy between antibody titer and therapeutic effect of BTK inhibitors, we focused on cytokines produced by B cells. Treatment with BTK inhibitors decreased pro-inflammatory cytokines such as IL-6 and IL-23 produced by topo I-reactive B cells (Figure 7, C and D). On the other hand, inhibitory cytokines such as IL-10 and IL-35 produced by topo I-reactive B cells increased (Figure 7, C and D). Analysis using anti-cytokine antibodies and cytokine knockout mice also supports these results (Figure 6, E and F). Based on the above, we hypothesized that cytokines produced by B cells are more important in the pathogenesis of skin sclerosis and ILD than antibodies themselves. The mechanism of action of the BTK inhibitor was thought to be that it improved the condition of SSc by inhibiting the production of inflammatory cytokines from B cells rather than by its effect on antibody production.

For more supportive data, we have recent data on the use of Rituximab (RTX; anti-CD20 antibody) in human SSc patients. Like BTK inhibitors, this treatment acts on B cells and eliminates almost all CD20-expressing B cells. In this trial, skin sclerosis and ILD were significantly improved by RTX treatment compared to the placebo group. However, the serum antibody titer remained consistently the same in both the actual drug group and the placebo group (Ebata, S, et al. Lancet rheum, 3; E489-E497, 2021. Supplement Figure 8).

In a new study of 10 consenting patients from a different population, data from one year after RTX showed an average improvement of about 6 points in mRSS, while titer remained the same (newly added Figure 7—figure supplement 1).

These results suggest that it is less the production of antibodies from B cells that is important in the pathogenesis of SSc, such as skin sclerosis and ILD, but rather the cytokines produced by B cells and the effects of B cells on T cells in the micro. As for the mechanism of therapeutic effect of BTK inhibitors, they did not change the antibody titer in serum, but suppressed inflammatory cytokine production, suggesting that they improve the condition by suppressing inflammatory cytokines.

The above information has been added to the text and Figure and Figure Legend.